# Carbonic Anhydrase Activators for Neurodegeneration: An Overview

**DOI:** 10.3390/molecules27082544

**Published:** 2022-04-14

**Authors:** Valeria Poggetti, Silvia Salerno, Emma Baglini, Elisabetta Barresi, Federico Da Settimo, Sabrina Taliani

**Affiliations:** Department of Pharmacy, University of Pisa, via Bonanno 6, 56126 Pisa, Italy; valeria.poggetti@phd.unipi.it (V.P.); emma.baglini@phd.unipi.it (E.B.); elisabetta.barresi@unipi.it (E.B.); federico.dasettimo@unipi.it (F.D.S.); sabrina.taliani@unipi.it (S.T.)

**Keywords:** carbonic anhydrase activators, neurodegenerative diseases, cognition enhancement, proton transfer, small molecules, basic moiety

## Abstract

Carbonic anhydrases (CAs) are a family of ubiquitous metal enzymes catalyzing the reversible conversion of CO_2_ and H_2_O to HCO_3_^−^ with the release of a proton. They play an important role in pH regulation and in the balance of body fluids and are involved in several functions such as homeostasis regulation and cellular respiration. For these reasons, they have been studied as targets for the development of agents for treating several pathologies. CA inhibitors have been used in therapy for a long time, especially as diuretics and for the treatment of glaucoma, and are being investigated for application in other pathologies including obesity, cancer, and epilepsy. On the contrary, CAs activators are still poorly studied. They are proposed to act as additional (other than histidine) proton shuttles in the rate-limiting step of the CA catalytic cycle, which is the generation of the active hydroxylated enzyme. Recent studies highlight the involvement of CAs activation in brain processes essential for the transmission of neuronal signals, suggesting CAs activation might represent a potential therapeutic approach for the treatment of Alzheimer’s disease and other conditions characterized by memory impairment and cognitive problems. Actually, some compounds able to activate CAs have been identified and proposed to potentially resolve problems related to neurodegeneration. This review reports on the primary literature regarding the potential of CA activators for treating neurodegeneration-related diseases.

## 1. Introduction

Carbonic anhydrases (CAs) are a superfamily of metalloenzymes that mainly catalyze the interconversion between CO_2_ and bicarbonate (CO_2_ + H_2_O ⇋ HCO_3_^−^ + H^+^) by using a metal hydroxide nucleophilic mechanism [1]. 

These enzymes are ubiquitous and exist both in eukaryotic and prokaryotic organisms. Coding genetic families have a different evolution, and they are: α-CA (vertebrates, bacteria, algae, cytoplasm of green plants), β-CA (bacteria, algae, chloroplasts of dicotyledons and monocotyledons), γ-CA (archaea and other bacteria), δ-CA (some marine diatoms), ζ-CA (some marine diatoms), η-CA (*Plasmodium* spp.), θ-CA (marine diatom *Phaeodactylum tricornutum*), and ι-CA (diatom *Thalassiosira pseudonana*, algae, bacteria, and archaea) [2,3,4,5]. 

In vertebrates we can find α-CAs and, to date, in human beings, 15 isoforms have been identified with different catalytic activities, subcellular localizations, and tissue distributions: hCA I, II, III, VII, and XIII are cytosolic; hCA IV, IX, XII, and XIV are membrane-bound; hCA VA, and VB are mitochondrial; hCA VI is secreted in saliva and tears; and hCA IX, XII, and XIV are transmembranal. hCA VIII, X, and XI are acatalytic and preponderantly expressed within the brain [3].

The main differences among these isoforms of α-CA are associated with secondary and tertiary organization of the apoprotein, which delineate physical/chemical characteristics, but the active site remains almost unchanged [4,6]. 

Since CAs catalyze the hydration reaction of CO_2_, they are involved in several physiological processes related to cellular respiration and to the transfer of CO_2_ with HCO_3_^−^ through tissues of metabolization and lungs. They also regulate pH levels; CO_2_ homeostasis; the secretion of electrolytes in several tissues and organs; a number of biosynthesis processes such as gluconeogenesis, lipogenesis, and ureagenesis; bone resorption; and calcification [3].

In addition to physiological functions, CAs are also involved in pathogenetic processes such as carcinogenesis, obesity, and epilepsy; for this reason, they are considered valid targets to develop potential drugs for the treatment or prevention of these pathologies [2,7].

The active site of α-CAs is a hydrophobic pocket 15 Å in depth, constituted by three main components (Figure 1): the metallic (II) ion arranged in a tetrahedral geometry, which is represented by the Zn^2+^ ion situated at the bottom of the cavity; three histidine residues, His94, His96, and His119, that bind the metal ion thanks to their imidazole moiety; and a water molecule in the inactive state of the enzyme, which loses a proton to produce the hydroxylated state of the enzyme, which represents the active form [8]. 

The water molecule bound to Zn(II) is also involved in a hydrogen bond interaction with the oxygen of the hydroxyl group of Thr199, which is in turn engaged in another hydrogen bond interaction with the oxygen of the carboxylate group of Glu106 (Figure 2). 

These interactions direct the substrate, represented by CO_2_, in a favourable position for the nucleophilic attack by the zinc-bound hydroxide, deriving from water, which loses a proton and increases its nucleophilicity [10].

The inactive state of the enzyme is constituted by a water molecule bound to the Zn (II) (Figure 3D). In the first step, the water molecule loses a proton and transforms itself into a hydroxylated species (Figure 3A), which is the active one. The hydroxyl ion acts as a strong nucleophile on the CO_2_ molecule (Figure 3B), which is situated in a nearby hydrophobic pocket, leading to the formation of a species in which the bicarbonate ion is coordinated to the metal ion (Figure 3C). Subsequently, the bicarbonate ion is displaced from the Zn (II) and released in solution by an incoming water molecule, which rebuilds the inactive and acid form of the enzyme (Figure 3D), and the catalytic cycle can restart [11]. The rate-limiting step in the activation process is the proton transfer from the water molecule bound to the Zn (II) to the surrounding environment, namely the proton shuttling. In the native state of the enzyme, this reaction takes place thanks to the imidazole and basic moiety of His64 (mainly in hCA I, II, IV, V, VII, and IX), which behaves like a proton shuttle because it has a pKa of around 7 [11].

The catalytic cycle of the enzyme can be represented by these reactions [2]:EZn^2+^—OH^−^ + CO_2_ ⇆ EZn^2+^—HCO_3_^−^ ⇆ EZn^2+^— OH_2_ + HCO_3_^−^
(1)
EZn^2+^—OH_2_ ⇆ EZn^2+^—OH^−^ + H^+^
(2)

Carbonic anhydrases activators (CAAs) are compounds able to behave as additional shuttles towards the His residue, transferring the proton and activating CAs even more.

### 1.1. Carbonic Anhydrases Activation 

CAs activation is a little-explored field compared to CAs inhibition. While CAs inhibitors (CAIs) have been in clinical use for many decades and are already used in therapy for the treatment of some pathologies such as glaucoma [12], epilepsy [13], obesity [14], CAAs, to date, have not shown relevant pharmacological applications and are still under study. 

The general mechanism that leads to CAs activation was suggested in 1990 and it can be schematized by the following reactions [3]: EZn^2+^—OH_2_ + A ⇆ [EZn^2+^—OH_2_—A](3)
[EZn^2+^—OH_2_—A] ⇆ [EZn^2+^—HO^−^—AH^+^](4)
[EZn^2+^—HO^−^—AH^+^] ⇆ EZn^2+^—HO^−^ + AH^+^
(5)

The activator binds to the enzyme in the active site and then, together, they constitute the enzyme–activator complex. Kinetic analysis demonstrated that the activator, once bound to the enzyme, does not compromise the affinity for the substrate (K_M_), having an impact only on K_cat_. Studies on Co (II)-substituted CAs complexed with the activator molecule established that there are no changes in the electronic spectra of complexes, similarly to the pure enzyme, highlighting that the activator does not bind to the metal ion, but to a different site. X-ray crystallographic data evidenced His64, fundamental for its proton-shuttling role, in two different orientations in the hCA II active site, far away from the Zn ion. The first is called “in” conformation in which His64 points toward the metal ion, while the second is named “out” conformation in which His64 points toward the exit of the active site. X-ray crystallographic data, related to the CAA histamine, also demonstrated that, while CAIs bind to the Zn (II), CAAs bind at the entrance of the cavity of the active site, far away from the metal ion. Moreover, activator’s binding site does not overlap with the region in which His64 is located [3].

The CAA is anchored to Gln92 thanks to H-bonds on the side chains of Asn62, Asn67, and Gln92, and to the water molecule, and this leads to a rearrangement of the H-bond network in the active site. In this way, the activator is ready to participate in the rate-limiting step of the catalytic cycle, working as an additional proton shuttle towards His64. In other words, together the activator molecule and His64 lead to the formation of the nucleophilic and active species of the enzyme in which a hydroxyl ion is coordinated to the metal ion instead of the water molecule [8]. 

### 1.2. Potential Therapeutic Applications of CAAs 

Recent findings reported an involvement of CAs in cognitive and memory disorders, suggesting that the activation of these enzymes may represent a potential effective strategy for the strengthening of synaptic efficacy [15,16] and the development of agents useful for the treatment of neurodegenerative pathologies such as Alzheimer’s disease. 

In particular, cerebral isoforms of CAs (I, II, IV, VA, VII, IX, XII, and XIV) may represent a target for an unexplored field to develop new drugs in psychiatric or neurodegenerative disorders. 

This hypothesis arises from the observation that CAs activation increases spatial memory in rats [15]. The administration of acetazolamide (a well-known non-selective CAI) to CD1 mice reduces CAs activity in the brain and causes amnesia in recognizing objects (OR), while the treatment with D-phenylalanine (CAA) that unselectively increases CAs activity, strengthens OR memory by activating extracellular signal-regulated kinase (ERK) [17]. 

Sun and Alkon reported that following the administration of phenylalanine to experimental animals, a remarkable pharmacological improvement of synaptic efficacy, spatial learning, and memory, was produced, demonstrating that CAAs could be used for managing particular conditions in which learning and memory are compromised [15,18]. 

In addition, it has been proven that the levels of different cerebral isoforms of α-CA (mainly CA I and II) are significantly reduced in patients with Alzheimer’s disease. 

Other studies revealed that CA I and II resulted in being oxidized and having a reduced catalytic activity in the frontal cortex and hippocampus of patients with Alzheimer’s disease. Consequently, the CA I and II catalytic activity dysfunction leads to imbalances of intracellular and extracellular pH levels, triggering protein aggregation and contributing to the progression of the disease [19,20]. Very recently, also the idea of potentially repurposing CAIs for the prevention of cerebrovascular and neurovascular pathology in AD and stroke was proposed [21].

Agents able to repristinate the catalytic activity of these isoforms or to increase other cerebral isoforms (such as CA VA, CA VB, and CA VII) could represent innovative approaches for the treatment of Alzheimer’s disease. 

As discussed above, CAs play an important role in several physiological functions, including the regulation of pH levels in the neurons and in the extracellular space through the regulation of ionic gradients of bicarbonate ion. Specifically, CAs contribute to the availability of protons and bicarbonate ions, which is necessary for the transmission of neuronal signalling. This in turn influences the purpose of proton-sensitive membrane proteins, regulating the kinetics and the concentration of pH transition into intra- and extracellular compartments [22,23]. These proteins are gamma aminobutyric acid agonist receptors (GABA_A_Rs) [24], *N*-methyl-D-aspartate receptors (NDMA), and ionic channels [25].

It has been demonstrated that into rats’ hippocampus, the excitation mediated by GABA_A_R depends on HCO_3_^−^ concentration, which is regulated by the cytosolic activity of CAs and suppressed by inhibitors [26]. 

Information processing and memory storage need a synchronized neuronal activity, commonly known as hippocampal theta rhythm, which was demonstrated to be associated to the GABAergic postsynaptic depolarization into pyramidal cells of adult rats’ hippocampus, with an inverted potential from Cl^−^ to HCO_3_^−^, and this process is regulated by CAs. Moreover, it has been shown that theta rhythm is inhibited also by CAs inhibitors and that CAs inhibition compromises rats’ spatial learning, but it does not influence other sensory or locomotor behaviours. This evidence implicates that CAs activity has effects both on theta rhythm and on memory consolidation through the signalling transmission mediated by HCO_3_^−^ [26]. 

Moreover, there are two putative mechanisms underlying CAs actions on cognition. The first regards GABA and the second regards ERK pathways. 

CAAs are reported to increase the efficacy of temporal activity of cholinergic and GABAergic inputs, transforming the hyperpolarizing GABAergic postsynaptic potential (PSP) from inhibitory (IPSP) to excitatory (EPSP), because they reduce intracellular concentrations of HCO_3_^−^, favouring its outflow through the channel receptor GABA_A_ (Figure 4A). Thus, the regulation of ionic gradients has several effects on post-synaptic depolarization with benefits of increasing memory and learning [27]. Becoming excitatory, the GABAergic postsynaptic potential potentiates the signal transfer thanks to the activity of CAs, which act as a gate. 

On the other hand, a study conducted on CD1 mice demonstrated that D-phenylalanine increases learning thanks to the activation of extracellular pathways of extracellular-signal-regulated kinase (ERK) into the cortex and hippocampus. This kinase is involved in the building of memory, in both the cortex and hippocampus, which are two very important cerebral areas for memory elaboration, especially for long-term memory (Figure 4B). In fact, the activation of ERK pathways in these two cerebral districts leads to an increased genomic response and memory encoding due to structural synaptic variations [17].

In a recent study, P. Blandina et al. (2020) [28] demonstrated, following the treatment with D-phenylalanine, an increased phosphorylation of ERK (1 and 2) in hippocampal and cortical cell homogenates, and these results are in agreement with previous studies showing that ERK phosphorylation in the amygdala was also inhibited by the treatment with acetazolamide [29]. 

After phosphorylation, ERK pathway induces a genomic response, which is very important for the consolidation of long-term memory. Indeed, this is related to the transformation of short-term learning into long-lasting memory in which CAs play a pivotal role. This study also demonstrated that the administration of acetazolamide causes amnesia during a non-spatial memory test, confirming that CAs are modulators of learning and memory [17]. Based on these findings, a number of CAAs are under study as potential candidates for the development of new drugs for the treatment of neurodegenerative diseases, in particular those related to memory disorders [17].

Actually, even if the hydration reaction of CO_2_ can occur without a catalyst, anyhow it is too slow at physiological levels of pH and it cannot satisfy metabolic demand, making the catalytic activity of CAs indispensable. The catalytic cycle mentioned above demonstrates that the active state of the enzyme is the basic and hydroxylated one, and it derives from the inactive and acid form. The transfer of the proton from the water molecule to the surrounding environment is the rate-limiting step of the entire process of activation and it takes place thanks to a basic group (imidazole moiety of His64). From this evidence, it was demonstrated that a molecule able to participate in the proton transfer, acting as an additional proton shuttle towards His64, may contribute to activate the enzyme. In this respect, CAAs must feature two main structural requirements: be a small molecule suitable for the active site, and possess at least a protonatable basic moiety with pKa values in the range of 6.5–8.0 for participating in the proton transfer [4]. 

In addition, also tissue engineering is a field in which CAs activation takes place. As already known, carbonate deposition in animals such as mollusk shells depends on bicarbonate formation catalyzed by CAs [30,31]. 

In fact, Muller et al., studying human osteogenic SaOS-2 cells, demonstrated that following their exposition to calcium bicarbonate in vitro, there was an increased Ca-deposit formation and also an amplified upregulation of hCA II gene expression [31]. 

Calcium carbonate formation is an important step in the biomineralization and bone formation processes since it behaves as a bioseed for the precipitation of calcium phosphate, and it has been shown that in sponges the presence of CAAs leads to an increased formation of the first one [30]. These findings paved the way to an additional application of CAA that resides in the bone formation process. 

## 2. Carbonic Anhydrase Activators (CAAs)

In this chapter, the activators of carbonic anhydrase (CAAs) reported so far in the literature are described and classified, focusing the attention on their chemical structures, updating previously published reviews [3,4]. 

Of note, CAAs biological profiles on the various CA isoforms are discussed with particular attention to those compounds able to behave as effective CAAs in vitro and that therefore could be interesting for animal studies regarding their involvement in cognitive processes. When possible, structure–activity relationships (SARs) are also discussed.

### 2.1. Amino Acids and Amines

Since an important structural requirement to activate CAs is a basic protonatable moiety as a mimic of the proton shuttling residue, the first molecules studied as CAAs were amino acids and their related amines.

As already discussed, the first X-ray crystallographic data on the complex between hCA II and the activator histamine [32] demonstrated and confirmed the theory that the activator-binding site is situated far away from the Zn(II) ion, reaching the edge of the cavity of the active site in the same region occupied by the two conformations of His64 [3]. 

In this regard, other compounds such as D- and L-phenylalanine [33], D- and L-histidine [34], D-tryptophan [35], and L-adrenaline [36] have been studied as activators, and they all bound to hCA II in a region called activator-binding site A [3], which is located on the opposite part of the region occupied by His64. 

There is only one exception that regards D-Trp, which, in addition to activate binding site A, also binds to a region called activator-binding site B, which is further out with respect to A [35]. 

Subsequently, several amino acids and also some related natural and synthetic amines have been studied [37] as CA activators, and the most interesting data were collected and reviewed by Supuran in 2018 [3] and by Angeli et al., in 2021 [4]. In particular, amino acid compounds **1**–**6** and also some amines **7**–**14** (Table 1) have been tested on 13 catalytically active human CAs isoforms: hCA I–VII, IX, and XII–XV, and their activation data were expressed as activation constants (K_A_).

In Table 1, the most interesting results are reported.

From a general point of view, all compounds **1**–**14** activate all tested hCA isoforms with diverse profiles showing K_A_ values ranging from the low nanomolar (9.0 nM of **12** for hCA IX) to the high micromolar (86 μM of D-Phe **1** for hCA I) range.

Some isoforms are better activated by amino acids, rather than amines. For example, hCA I is better activated by D-Phe **1**, D-His and L-His **2** with K_A_ 30–90 nM, compared to histamine **7**, which activates hCA I with K_A_ 2.1 μM [3]. 

Moreover, some amino acids are characterized by a high eudysmic ratio, for example L-Phe **1** is more active than D-Phe **1** on hCA I with K_A_ values of 70 nM and 86 μM, respectively. Conversely, D-Phe **1** is more active on hCA VB (K_A_ 70 nM) and on hCA XIII (K_A_ 50 nM) with respect to L-Phe **1** (K_A_ values of 10.45 μM and 1.02 μM, respectively). 

hCA XIII isoform is also well activated by D-Trp **4** with K_A_ 0.81 μM and by D-DOPA **3** with K_A_ 0.81 μM, while L-DOPA **3** is endowed with a great affinity for hCA VA and VB (K_A_ values of 36 nM and 63 nM, respectively) [4]. 

This reveals that also small structural differences comprising the enantiomeric form can lead to a more or less strong activation. In fact, after binding with the enzyme, the activator interacts in several ways and with different amino acids residues and water molecules of the binding site. These interactions can be both favorable and unfavorable depending on the residues involved and this leads to a better or a worse activation of the different isoforms [3]. 

Interestingly, D- and L-His **2** are the only amino acids that strongly activate hCA VII (K_A_ values of 0.71 μM and 0.92 μM, respectively), the predominantly cerebral isoform that is activated also by D- and L-DOPA **3** and neurotransmitters such as dopamine **8** and serotonin **9**. These neurotransmitters, as well as histamine **7**, showed also to potently activate the mitochondrial isoforms, hCA VA, which are also present in the nervous tissues [4]. 

Some biogenic amines maintain the same activating features of their amino acidic precursors (serotonine—Trp), but some others change them (histamine—His). This happens because for some amino acids like His, the carboxylic group is necessary for strengthening the ability to activate CAs [4]. 

From the analysis of the data reported so far for compounds **1**–**14** [3,4], it is clear that none of the amino acids **1**–**6** or amines **7**–**14** are isoform-selective CAA.

Anyhow, a certain selectivity is observed for example for histamine **7**, which shows a K_A_ value of 10 nM towards the mitochondrial isoform hCA VA and the transmembrane one hCA XIV, while it results as a moderate activator (K_A_ values in the micromolar range) for all the other CA isoforms.

The most interesting compound from the point of view of neurodegeneration, however, is histidine **2**, which activates the isoforms most expressed in the brain, namely hCA I, VA, and VII.

### 2.2. Histamine-Based Compounds

Histamine **7** was one of the first CAAs that was studied, and it was used as a lead compound for the development of novel and more potent CAAs. 

Over the years, several modifications have been made to the structure of histamine for the development of CAAs.

These changes can be schematically classified as follows: replacement of the imidazole ring with other heterocycles, derivatization of the primary amino group, insertion of halogens, and insertion of another imidazole ring. Subsequently, several classes of histamine-inspired compounds were developed. 

#### 2.2.1. Replacement of the Imidazole Ring

In one of the first studies conducted by Supuran et al. in 1996, the imidazole ring of histamine was replaced with several heterocycles like 2,4,6-trisubstitued pyridinium (**15**), 1,3,4-thiazole (**16**), or both (**17**) (Table 2) [38]. 

The compounds of series **15**–**17** showed to be efficient hCA II activators at 10 μM concentration, displaying activation rates between 147 and 184%. 

#### 2.2.2. Derivatization of the Primary Amino Group

In the following years, some X-ray crystallographic studies demonstrated that the amino group of histamine does not interact with the enzyme, so this moiety was substituted with different groups in order to obtain compounds such as carboxamides **18**, ureas or thioureas **19** [39], sulfonamides **20**, arylsulfonylureido derivatives **21** [40], and histamine dimers comprising polyaminoacyl moieties like EDTA **22** [39] and aminoacyl histamines **23** [41] (Table 2). All these compounds showed to efficiently activate hCA I, II (h = human), and bCA IV (b = bovine), with K_A_ values in the low nanomolar-low micromolar range, as reported in Table 2 [4]. 

In particular, for compounds **18**, **19**, and **22**, an efficient activation was observed towards all three isozymes, but especially for hCA I and bCA IV, with the best performing compounds displaying K_A_ values in the nanomolar range. Anyhow, some of them also showed to activate hCA II with K_A_ values around 10–25 nM [39]. The same trend was also observed for aminoacyl histamines **23** [41].

Sulfonamides **20** activated all three isozymes. The hCA I showed to be particularly sensitive (K_A_ 6.0 nM–0.28 μM). The arylsulfonylureido derivatives **21** were the best CA activators of these series, with K_A_ values in the 3–6 nM range for hCA I, 8–150 nM for hCA II, and 10–30 nM for bCA IV, thus being 1500 times stronger hCA II activators compared to histamine, although they are non-selective [40]. 

These series of CAAs might lead to the development of drugs/diagnostic agents for genetic disease of bone, kidneys, and above all, brain, since they mainly activate the CA isoforms most expressed in these compartments (namely hCA I, hCA II, and hCA VII) [39,40,41].

In these studies, it was shown that the derivatization of the aminoethyl moiety that does not interact with the enzyme is useful to make it point outwards to the active site or to increase the stabilization of the activator–enzyme complex. Based on the earlier study on the ureido histamine derivatives [39], Licsandru et al. [11] decorated the aminoethyl chain of histamine and synthetized ureido **19** (X = O) and bis-ureido **24** (Table 2) derivatives featuring alkylof different length, cycloalkyl, and aryl moieties at R, R′. 

Both new ureido- **19** (X = O) and diureido-derivative **24** do not activate hCA II as their K_A_ values are > 200 μM, but they moderately activate hCA I with K_A_ values ranging between 0.73 and 3.4 μM. In particular, the best hCA I activator **24a** (Table 2) (K_A_ 0.73 μM, 2.74 times more active than histamine) belongs to diureido derivatives of series **24** [11]. 

In general, ureido derivatives are worse hCA I activators than bisureido derivatives, and this is probably due to the long alkyl chain that is unfavorable for the binding of these compounds at the entrance of the active site due to its hydrophilic nature [1]. 

The data reported by Licsandru et al. [11] show a different ability of the compounds investigated to activate two distinct isoforms, hCA I and II. This undoubtedly depends on the nature of the entrance of the active site cavity: The one of hCA II is more hydrophilic than the hCA I one, which is characterized by the presence of six histidine residues. In this view, the lipophilic nature of compounds **19** (X = O) and **24** prevents the binding to the active site, and thus, the activation of the hCA II isoform while favouring that of hCA I.

In 2011, Dave et al. derivatized histamine with pyridinium salt to obtain compound **25** (Table 2), which was crystallized in complex with hCA II, showing remarkable binding (hCA II 156% activation rate at 20 μM) [42]. 

On the basis of this interesting study, compound **25** was further functionalized and new pyridinium histamine derivatives **26** (Table 2) were developed [43]. 

Compounds of general formula **26** were investigated as activators of three cytosolic isoforms, hCA I, II, and VII, which are widely expressed in the brain, and they displayed activities ranging from the subnanomolar to the micromolar range.

While histamine **7** acts towards hCA I as a low micromolar activator (K_A_ 2 μM), its substituted pyridinium derivatives **26** show a range of activities, with K_A_ values of 0.5–93 μM.

The physiologically dominant cytosolic isoform hCA II is weakly activated by histamine **7** (K_A_ 125 μM), whereas all compounds of series **26** showed to be better CAAs with K_A_ values in the range of 9–78 μM. Finally, hCA VII is activated by histamine (K_A_ 37.5 μM) and is also activated by all pyridinium salts **26**, but with a variable profile. Based on the substitution pattern at the pyridinium ring, it was possible to identify in the compounds **26** three groups of hCA VII activators. The 4-phenyl-2,6-disubstituted salts incorporating bulkier alkyl moieties and trialkyl-substituted derivatives were shown to be weak hCA VII activators (K_A_ 19–81 μM). The second group, with heterogeneous substitution patterns, such as the disubstituted one, and the trisubstituted ones with only alkyl moieties or incorporating the styryl moiety, showed better affinity for hCA VII (K_A_ 2.15–12.5 μM). The diethylpyridinium derivative, the trisubstituted ones (with only alkyl moieties at the pyridinium ring), and the tetrasubstituted pyridinium salts showed to be excellent hCA VII activators (K_A_ 0.8 nM–1.16 μM) [43].

In a more recent study, (hetero)aryl substituted thiazol-2,4-yl derivatives of general formula **27** (Table 2) were developed [44]. 

In particular, compound **28** was shown to be a good activator mainly towards isoforms hCA I, II, and VII, activating I and II isoforms with K_A_ values an order of magnitude greater (K_A_ 63.4 μM and 68.1 μM, respectively) than isoform VII (K_A_ 7.5 μM) [44]. 

#### 2.2.3. Insertion of Halogens

Two different studies [45,46] were conducted on histamine analogues in which halogens were inserted on the imidazole ring. The presence of halogens on the histamine structure leads to changes in its physico-chemical features, because of the withdrawing properties of halogens themselves.

Both mono- and di-halogenated derivatives (**29** and **30**, respectively, Table 3) were investigated, featuring chlorine, bromine, or iodine. Moreover, also *N*-*tert*-butyloxycarbonyl derivatives **31** and **32** (Table 3) were synthetized and tested as CAAs on isoforms hCA I and II. Boc-protected compounds showed to have a lower capacity to activate CAs than the deprotected ones. In fact, the first have K_A_ values in the range of 5.4–29.3 μM towards hCA I and of13.6–50.2 μM towards hCA II, and the second have K_A_ values in the range of0.7–21 nM for hCA I and of 1.0–115 nM for hCA II. The obtained data revealed also that the mono-halogenated derivatives are more active than the di-halogenated ones, and the activation of the enzyme increases with the growth of the molecular weight of the compounds [45]. 

#### 2.2.4. Insertion of Another Imidazole Ring

With respect to the insertion of another imidazole ring, in 2014, Draghici et al. developed compounds of general structure **33** (Table 3), in which two imidazole rings were linked by means of an ethyl chain and featured substituents at position 2, characterized by increasing steric hindrance (H, Me, Et, *i*-Pr, and Ph) [47]. 

More specifically, one of the two imidazoles behaves as an additional proton shuttle towards His64, and the other acts as a binding point to the CA-active site edge [51]. The presence of small groups such as methyl or hydrogen groups at position 2 of the imidazole result in very strong (nanomolar K_A_ values) and selective activators of the hCA VA (K_A_ 9.0–131 nM) and VII (K_A_ 15–89 nM) isoforms present in the human brain [47].

#### 2.2.5. Histamine-Inspired Compounds

Synthetic compounds **34**–**36** have been studied as activators too and included into the “histamine-inspired compounds” group, as well as natural compounds such as *L*-(+)-ergothioneine (**37**), melatonin (**38**), and spinacine (**39**) (Table 3) [48]. 

Both synthetic (**34**–**36**) and natural (**37**–**39**) derivatives were assayed as activators of four human isoforms of CA, namely hCA I, II, IV, and VII. Most of compounds **34**–**39** activated hCA I and VII in the micromolar range, with K_A_ values ranging between 0.12 µM and 34.7 µM, whereas they were shown to be weak activators of hCA IV (K_A_ 53.4–80.5 μM) and not active towards hCA II (K_A_ values > 100 µM). In this series, two natural compounds, *L*-(+)-ergothioneine (**37**) and melatonin (**38**), are noteworthy and deserve to be further developed as they were shown to be selective and quite potent hCA VII activators (K_A_ values of 820 nM and 120 nM respectively). Instead, spinacine (**39**) was shown to be the best hCA I activator with a K_A_ value of 7.21 μM that, unfortunately, is higher than the histamine one [48]. 

Subsequently, Akocak et al. synthetized spinaceamine-substituted compounds **40** (Table 3). In particular, 14 spinaceamine derivatives from this class were obtained and evaluated for their ability to activate hCA I, II, IV, and VII isoforms [49]. 

Spinaceamine is a bicyclic alkaloid present in the skin of Australian amphibians and structurally is considered a product that originates from the cyclization of histamine. It has got two units of protonatable nitrogen in its structure that in turn can participate in the rate-limiting step of the activation process [52]. 

Isoform hCA I is moderately activated by all the tested compounds **40** with K_A_ values in the range of 2.52–21.5 μM. These compounds also activate hCA II with K_A_ values in the range of 0.60–17.2 μM. As regards the hCA IV isoform, it is little activated by all compounds **40** (K_A_ 0.52–63.8 μM) 

Finally, the cerebral isoform hCA VII is potently and quite selectively activated by these derivatives; in fact, apart from a few compounds that display K_A_ greater than 1 μM, all the other tested compounds potently activate this isoform with nanomolar K_A_s (82–840 nM) [49]. 

Additionally, Akocak et al. worked on the modification of histamine by inserting a second imidazole ring and thus developed bis-spinaceamine-substituted derivatives (**41**) and bis-histamine Schiff bases (**42**) (Table 3) [50]. 

Three bis-spinaceamines **41a**,**b**,**d** and four Schiff bases **42a**–**d** were synthetized and tested as activators on hCA I, II, IV, and VII isoforms, as reported in Table 3. 

The data obtained evinced that hCA VII is the most sensitive isoform to be activated by all the tested compounds. In particular, the three bis-spinaceamines **41a**,**b**,**d** are the most interesting compounds since they can be considered as quite potent and selective activators (K_A_ values in the range of 32–39 nM), regardless of the spacer (X) that featured. Instead, bis-histamine Schiff bases **42a**–**d** display moderate (K_A_ values ranging from 3.28 to 42.1 μM) activity on all four isoforms with compound **42d** (furyl substituted), which selectively activates the hCA VII isoform (K_A_ 85 nM) [50]. 

### 2.3. Histidine- and Carnosine-Based Derivatives 

As well as histamine **7**, histidine **2** (Table 1) was also used as lead compound for the design of CAAs and one of the first changes was done on the primary amino moiety. In fact, in 2001, Scozzafava et al. modified histidine and its dipeptide with β-alanine, carnosine **43** (Table 4), and some structurally related dipeptides bearing basic amino acids such as arginine or lysine (R), thus obtaining arylsulfonylureido tri-/tetra-peptide derivatives of general formula **44** (Table 4) [53]. 

Many of the compounds **44** displayed K_A_ values in the 1–20 nM range for hCA I and bCA IV, and in the range of 10–40 nM for hCA II. Interestingly, ex vivo experiments revealed that some of them were able to strongly enhance cytosolic red cells CA activity in human erythrocytes, thus being able to behave as effective in vivo CAAs, and might thus constitute interesting candidates for animal studies regarding their involvement in cognitive processes [53]. 

In another study in 2009, Abdo et al. developed arylsulfonylhydrazido-L-histidine derivatives bearing a 4-substitued aryl moiety; these compounds were shown to be less active than histidine itself, except for **45** (Table 4), which activates hCA II with a K_A_ 0.21 μM [54]. 

Like histamine, also histidine methyl ester (**46**) and carnosine methyl ester (**47**) were derivatized, introducing electron-withdrawing halogens, and obtaining mono (X_2_ = H)- and di-halogenated (X_2_ = Cl, Br, I) analogues (Table 4) [46]. 

The obtained compounds of series **46** and **47** resulted in activating in a diverse manner the hCA I, II VII, XII, and XIV isoforms tested. The most interesting results are summarized in Table 4.

The differences in the activation properties of the synthetized compounds can be correlated both to the size of the halogens inserted into the molecular structure and their electron-withdrawing properties, as they modify the distribution of electrons into the ring and, in turn, the nucleophilicity and the basicity of the entire molecule that could interfere with the ability to activate the enzyme. 

From a general point of view, mono- and di-halogenated L-His-OMe derivatives **46** are more effective hCA I and II activators compared to the L-Car-OMe **47** counterparts, but are less effective than the corresponding histamine derivatives (series **29**–**30**, Table 3).

The best hCA I and II activator is compound **46a** (K_A_ 0.9 nM and 12 nM, respectively), which is a mono-halogenated Boc-substituted derivative. Its analogue with the deprotected amine (**46b**, K_A_ 1.2 nM and 0.32 μM, respectively) or the corresponding di-halogen derivative (**46c**, K_A_ 1.5 nM and 13 nM, respectively) are instead less effective activators of hCA I and II. This general trend is observed for all the compounds of the series.

The corresponding derivative L-Car-OMe of compound **46a**, namely **47a**, weakly activated hCAI and II (K_A_ 4.3 μM and 5.2 μM, respectively), while it showed to be an effective activator of isoform VII (K_A_ 0.81 μM), which is more effective than the corresponding L-His-OMe derivative **46d** (K_A_ 9.7 μM). In fact, in general, L-Car-OMe derivatives **47** proved to be effective activators of the cerebral isoform hCA VII, which is weakly activated by L-His-OMe derivatives **46** and also by the corresponding histamine derivatives (series **29**–**30**, Table 3).

The best-performing hCA VII activator is compound **47b** (K_A_ 8.1 nM), which is a di-halogenated derivative; the relative Boc- analogue **47c** is instead a less effective hCA VII activator (K_A_ 4.0 μM**)**. Thus, series **47** displayed an opposite trend with respect to series **46** for what concerns the activation of hCA I and II isoforms.

hCA XII and XIV are both little activated by the two series **46** and **47,** with values of K_A_, aside for very few exceptions, in the range of 0.83–64.7 μM for hCA XII and 0.92–40.8 μM for hCA XIV [46]. 

In 2020, Vistoli et al. [55] biologically evaluated in hCA I, II, VA, and IX, some dipeptides that contain carnosine **43** (Table 4) and histidine **2** (Table 1) of general formula **48** (Table 4). In fact, L-carnosine is found in excitable tissues together with other histidine-containing peptides such as carnicine and L-carnosinamide (derivatized on -COOH), L-homocarnosine and Gly-L-His (different lengths of amino terminal residues), and the N-acetyl-carnosine derivative [56]. 

In general, compounds **48** activate the four human isoforms hCA I, II, VA, and IX in the low to high micromolar range. In particular, concerning hCA I, they showed K_A_ values in the range of 16.6–80.4 μM, with D-carnosinamide (**48a**, Table 4) and carnicine (**48b**, Table 4) being the most active (K_A_ 16.6 μM) compounds for this isoform. Isoform hCA II was poorly activated (K_A_ > 76.6 μM) by all the compounds tested. Also, hCA VA was shown to be weakly activated (K_A_ 6.4–52.8 μM), and the most active compound for this isoform was shown to once again be carnicine **48b** (K_A_ 6.4 μM).

In the end, the hCA IX isoform was a little better activated (K_A_ 1.14–50.2 μM), with L-anserine **48c** (Table 4) (K_A_ 1.14 μM) being the best and most selective activator of this isoform [55]. 

### 2.4. Gold Nanoparticles of Histamine, Histidine- and Carnosine Derivatives

Gold nanoparticles coated with bioactive substances demonstrated to be very fascinating for innovative biomedical applications because they seem to improve the site-specific drug delivery.

In this view, histamine, L-carnosine-methyl ester, and L-His-methyl ester were conjugated with lipoic acid to obtain compounds **49**, **50,** and **51** respectively, (Table 5) [57].

Compounds **49**–**51** were biologically evaluated on isoforms hCA I, II, IV, VA, VII, and XIV in comparison with their starting compounds, namely, histamine **7** (Table 1), L-His **2** (Table 1), L-His methyl ester, L-carnosine **43** (Table 4), and L-carnosine methyl ester, and the obtained data are reported in Table 6. 

Compounds **49**–**51** showed to be very potent CAAs both in vitro (K_A_ values ranging from 1.0 to 9.0 nM) and ex-vivo (in whole blood experiments, with an increase of 200–280% of the CA activity) of all the CA isoforms tested.

This is the first example of enzyme activation with nanoparticles and may lead to interesting biomedical applications [57]. 

### 2.5. Sulfur, Selenium and Tellurium Containing Amines 

In 2019, Tanini et al. synthetized three different series of β-aminochalcogenides containing sulfur **52**, selenium **53**, and tellurium **54** (Table 5) and investigated the CA-activating properties of the derivatives obtained for hCA I, II, VA, and VII isoforms [58]. 

The amphetamine represents the lead compound for the development of these new chalcogen-containing compounds, based on the results of a previous study conducted in 2017 in which psychoactive compounds demonstrated to be strong CAAs [59]**.**

All the synthesized compounds of series **52**–**54** showed to be moderate activators of hCA I (K_A_ 7.7–43.7 μM), hCA VA (K_A_ 5.2–23.3 μM), and the brain-associated cytosolic CA isoform, hCA VII (K_A_ 78.9–44.9 μM), without distinction about the presence of S (**52**), Se (**53**), or Te (**54**); hCA II was not activated by these compounds (almost all K_A_ values > 100 μM) (Table 5).

In addition, the presence of Se and Te makes derivatives of series **53** and **54** antioxidant compounds able to inhibit the formation of ROS metabolites, thus preventing cellular stress and damage, which are characteristics of neurodegenerative diseases. More specifically, they were tested as mimics of glutathione peroxidase and the best compounds were the aminotellurides (**54**). For this reason, these compounds can be considered able to contrast the progression of neurodegenerative processes due to their dual action: inhibition of oxidative stress (thanks to tellurium and selenium properties) and strengthening of synapses and neuronal activity (thanks to CAs activation) [58]. 

### 2.6. Drug Repurposing 

Several drugs already used in therapy for other aims and carrying proton-shuttling groups in their structure such as timolol [60] and propranolol (β-blockers), fluoxetine, sertraline and citalopram [61] (serotonin reuptake inhibitors), sildenafil [62] (phosphodiesterase IV inhibitor), and amphetamine/methamphetamine derivates [59] (psychoactive substances), were studied as CAAs. 

These compounds feature a central heterocyclic scaffold, which is not directly correlated to histamine or histidine, functionalized with an amino group linked to an alkyl spacer.

Timolol **55** (Table 7) is an anti-hypertensive drug, belonging to the class of β-blocker featuring of an aromatic and heterocyclic moiety and a secondary and protonatable amino group. Following enzymatic studies, it has been proven that timolol activates hCA I and II (K_A_ values of 12 μM and 9.3 μM, respectively), establishing a ternary complex with the enzyme and the substrate. It binds at the entrance of the active site, where His64 is located, and in a different position compared to the substrate, as timolol binding does not impact on the affinity of the substrate to its target, only increasing *v_max_* in enzymatic kinetics [60]. 

More cutting-edge research is represented by the SSRIs: fluoxetine **56**, sertraline **57,** and citalopram **58** (Table 7) are drugs used in therapy for the treatment of depression, and they bear one protonatable group in their molecular structure that makes them potential CAAs. 

All three SSRIs **56**–**58** demonstrated to be able to dose-dependently activate hCAI and II, with similar activation rates as those of histamine **7** and phenylalanine **1** (Table 1). In particular, hCA I is better activated by histamine **7** (170% at 1.0 μM) and fluoxetine **56** (175% at 1.0 μM), instead of hCA II, which is better activated by citalopram **58** (170% at 1.0 μM), and phenylalanine **1** (175% at 1.0 μM). Sertraline is the weaker activator of both hCA I and hCA II (145% and 140%, respectively, at 1.0 μM). These data are important because they pave the way to a new therapeutic approach that is particularly direct against Alzheimer’s disease associated with major depression [61]. 

Sildenafil **59** (Table 7) is a phosphodiesterase-5 (PDE5) inhibitor used in therapy for the treatment of erectile dysfunction. Its structure is very bulky, but it bears a piperazine moiety similar to that of some CAAs, giving the hope that also sildenafil can activate CAs [62]. From this view, in 2009, Sildenafil **59** was tested on hCA I-XIV isoforms [62]. From the obtained data (some interesting ones reported in Table 7), sildenafil **59** mostly activates hCA I, VB and VI with K_A_ values of 1.08 μM, 6.54 μM, and 2.37 μM, respectively, while isoforms hCA III, IV, and VA are less activated with K_A_ values in the range of 13.4–16.8 μM. Isoforms hCA II, IX, XIII, and XIV are little activated by sildenafil with K_A_ in the range of 27.5–34.0 μM. In the end, hCA VII and XII are the least activated isoform by sildenafil with K_A_ about 73 μM [62]. 

Sildenafil **59** has been also tested in vivo on rats. Unfortunately, CAs activity was shown to be decreased compared to baseline levels and this could be related both to the production of the metabolite “desmethylsildenafil” and to the production of NO/nitrate following PDE5 inhibition. Indeed, the first could bind with its NH group to the zinc-bound water molecule in the same way as phenols, a class of CAIs [63], while the second behaves as potent inhibitor [64]. 

Many psychoactive substances possess the phenethylamine scaffold and the general formula Ar–CH_2_CH(R)NHR’, and, as reported, this type of amine could effectively activate CAs [32,35].

In this contest, in 2017, five amines structurally related to amphetamine, namely amphetamine **60**, methamphetamine **61**, phentermine **62**, mephentermine **63,** and chlorphenteramine **64** (Table 7) were the first psychoactive substances to be investigated as CAAs and tested on 11 hCA isoforms (hCA I, II, IV, VA, VB, VI, VII, IX, XII, XIII, and XIV) [59]. 

Compounds **60**–**64** are used in therapy for treating attention deficit hyperactivity disorder, narcolepsy, obesity, and nasal congestion, but unfortunately, they have a lot of side effects including psychosis [65]. 

The data obtained (some interesting ones reported in Table 7) evinced that compounds **60**–**64** cannot be considered CAAs of the hCA I and II isoforms (K_A_ > 150 μM), while they moderately activate hCA XIII and XIV (K_A_ ranges 24.1–79.5 μM and 6.81–18.1 μM, respectively). Isoform hCA IX resulted not significantly activated by the amines investigated in this study, while the other tumor-associated isoform hCA XII demonstrated to be significantly activated by amphetamine **60**, methamphetamine **61,** and chlorphenteramine **64** with K_A_ values in the range of 0.64–0.94 μM, but less activated by the other amines (K_A_ range 3.24–6.12 μM). The most activated isoform is hCA IV (K_A_ values in the range of 51 nM–1.03 μM). In particular, the best hCA IV activators are methamphetamine **61** and chlorphenteramine **64** with K_A_ values of 51 nM and 55 nM, respectively, followed by phentermine **62** and amphetamine **60** with K_A_ values of 74 nM and 94 nM, respectively. Isoforms hCA VA and VB are effectively activated by compounds **60**–**64** with K_A_ values in the range of 0.24–2.56 μM and, particularly, the best activator of hCA VA showed to be chlorphenteramine **64** with a K_A_ 310 nM; the best activator of hCA VB demonstrated to be mephentermine **63** with a K_A_ 240 nM. 

Of note, the cerebral isoform hCA VII was successfully activated by all five compounds with K_A_ values in the range of 98–0.93 μM, with chlorphenteramine **64** being the best hCA VII activator of this series (K_A_ 98 nM).

Since some CAs (hCA VII, VA, VB, and XII) are abundant in the brain, some of the cognitive effects of these psychoactive compounds might be related to the activation of these enzymes, bringing new light to the intricate relationship between CA activation by these substances and their multitude of pharmacologic activities [59]. 

Very recently, the CAs activating effects of a series of histamine receptor (H_1_, H_2_, H_3,_ and H_4_) agonists/antagonists towards four hCA isoforms expressed in the human brain, namely hCA I, II, IV, and VII, were investigated [66]. 

In particular, 30 compounds (including histamine itself **7**, Table 1, and many well-known drugs, for example pyrilamine and loratadine among the H_1_ antagonists, and cimetidine and ranitidine among the H_2_ antagonists) were tested, and they were all shown to be moderate activators of hCA I (K_A_ range from 52 nM to >100 μM), hCA II (K_A_ range from 82 nM to >100 μM), hCA IV (K_A_ range from 1.02 μM to >100 μM), and hCA VII (K_A_ range from 110 nM to >100 μM, with hCA IV being the worst activated isoform. The most interesting results are reported in Table 8. In general, the best activators of hCA I isoform were the methyl histamine/histidines **65**, **66**, **67** (K_A_ range from 52 μM to 0.36 μM), the thiazole derivative **68** (K_A_ 0.87 μM), burimamide **69** (K_A_ 0.88 μM), metiamide **70** (K_A_ 0.98 μM), and impromidine **71** (K_A_ 0.72 μM), which are structurally related to histamine as they all bear the imidazole ring in their structure, except for compound **68,** which has a thiazole ring. In particular, 1-methyl histidine **67** has an additional proton transfer group represented by the carboxyl moiety, thus resulting in being much more potent (K_A_ 52 nM).

Moreover, some compounds such as α-methyl histamine **65** and 1-methyl histidine **67** were shown to strongly activate the hCA II isoform with K_A_ values of 82 nM and 0.57 μM, respectively.

Interestingly, hCA VII showed to be the most activated isoform, and impromidine **71** is the most active derivative (K_A_ 0.10 μM), followed by other compounds ranging between 0.11 and 7.05 μM. In particular, some interesting SARs were observed for this isoform, resulting from the presence of more lipophilic groups (as in **68**, **72**, **73** or **74**) able to promote a greater and more selective hCA VII activation [66]. 

### 2.7. Miscellaneous

#### 2.7.1. Ureas and di-Ureas Incorporating 1,2,4-triazole Derivatives

In 2017, Le Duc et al. substituted the imidazole moiety of a histamine with 1,2,4-triazole, developing two different series represented by mono-urea **75** and di-urea **76** compounds (Table 9) [67].

Fourteen derivatives were synthetized, in which R and R′ can both be alkyl and aryl groups, introduced for modulating the linker-length between the two triazole moieties (in series **76**) or for modifying lipophilicity and, in turn, the activity of the enzyme (in series **75**). 

Both series bear an ureido moiety, as it has been proven that this group gives flexibility to the various compounds to better accommodate at the entrance of the CA-active site cavity [69]. 

Mono-ureas **75** and di-ureas **76** were biologically evaluated on hCA I and II isoforms and they demonstrated to strongly activate hCA I with K_A_ values in the range of 0.81–993 nM. The best effective hCA I activators belong to series **76** and are compounds **76a** (K_A_ 0.81 nM) and **76b** (K_A_ 0.94 nM) (Table 9). Whereas the best hCA I activator of series **75** is compound **75a** (K_A_ 6.1 nM). Moreover, for series **75,** it was proven that the activity for hCA I diminishes with the increasing of the chain-length; SARs are instead more complex for series **76**. In general, regarding hCA I activation properties, K_A_ values are influenced by the length of the chain for series **75** and by the linker between the two rings for series **76** [67]. 

Even the hCA II isoform results in being well activated by the compounds of the two series **75** and **76,** with K_A_ values in the range of 0.05–6.7 μM. In particular, in series **75**, the best hCA II activator is compound **75a** (K_A_ = 1.7 nM), as well as for hCA I (K_A_ 6.1 nM), whereas compounds that activate the most hCA II belong to series **76**. The best hCA II activator showed to be **76b** (K_A_ 0.05 nM), which also strongly activates hCA I (K_A_ 0.94 nM), resulting in a very potent dual hCA I/II activator. In this series, also compound **76c**, with the 1,4-phenylene linker and the amino group, well activated the hCA II isoform (K_A_ 0.12 nM). 

Interestingly, compound **76b** is the most effective hCA II activator (K_A_ 0.05 nM) ever reported, and has shown to be a lead compound for the development of clinical candidates in cognitive impairment or other diseases characterized by a deficit of hCA I and hCA II isoforms, such as Alzheimer’s disease or aging [67]. 

#### 2.7.2. Amino Alcohol Oxime Ethers 

In 2021, Nocentini et al. [68] based their research on a previous study, in which it was demonstrated that the β-blocker timolol **55** (Table 7) was able to activate CAs. More specifically, this drug proved to bind at the entrance of the active site cavity and to activate hCA I and II isoforms by establishing a ternary complex with the enzyme and CO_2_ [60]. Thus, timolol **55** was identified as a lead compound for the design of a series of new activators. 

In this context, Nocentini et al. studied a series of amino alcohol oxime ether derivatives with a general structure **77** (Table 9), already known as β-blockers, analgesics, and antiarrhythmics, and tested their activation ability towards hCA I, II, IV, and VII [68]. 

All amino alcohol derivatives of series **77** showed to be good activators of the tested CAs, with K_A_ values spanning from low micromolar to nanomolar range. Specifically, hCA II and VII are most potently activated by these derivatives (K_A_ range of 79–420 nM) with respect to hCA I and IV (K_A_ range from 10- to 100-fold higher), with hCA IV being the less-activated isoform (K_A_ 1.01–12.9 μM).

In particular, compounds belonging to the tert-butylamino series turned out to better activate isoforms hCA I and II compared to those belonging to the iso-propylamino one. In fact, tert-butylamine **77a** (Table 9) is the most potent and selective hCA II activator (K_A_ 79 nM) of this study, while the iso-propylamines **77b** and **77c** (Table 9) showed potent hCA VII activation profiles (K_A_ 82 nM and 91 nM, respectively) and 10- to 100-fold selectivity with respect to the other CA isoforms tested [68].

#### 2.7.3. Imidazoline and Other Related Five-Membered N-heterocycle Derivatives

Imidazoline ring is considered a bioisoster of the imidazole moiety and it is present in the antihypertensive agent clonidine **78** (Table 10). In 2020, Chiaramonte et al. developed a series of 2-aminoimidazolines **79** (Table 10), structurally related to clonidine **78,** which showed to be able to activate several hCA isoforms (hCA I, IV, VA, VII, IX, XII, and XIII) with potency in the micromolar range, while it was inactive on hCA II. In particular, clonidine showed to best activate hCA VII and XIII isoforms with K_A_ values of 8.4 μM and 7.8 μM, respectively [70]. 

The 2-aminoimidazoline derivatives **79** were tested on five hCA isoforms (I, II, VA, VII, and XIII) and, as clonidine, they showed to be inactive on hCA II, while resulting good activators of hCA I, VA, VII and XIII isoforms (Table 10), with K_A_ values in the micromolar range.

In general, from the data obtained, it is evinced that most of the derivatives of series **79** were better activators than clonidine on all hCA isoforms tested. Moreover, some structure–activity relationships (SAR) were derived for these compounds that can be applied all over the hCA isoforms: the substitution with methyl at N-1 (R_1_ = CH_3_) improves the performance of compounds with respect to the unsubstituted ones (R_1_ = H), as well as the alkyl aryl chain compared to the alkyl one (R_2_). Finally, the amino or methylamino group is preferable to S as a linker (X) between the heterocycle and the alkyl-aryl chain. In addition, some other SAR were obtained and showed to be different on the various isoforms, suggesting that, in series **79**, it could be possible to develop new compounds more selective for a particular CA isoform. In particular, the most interesting compounds are **79a** and **79b**, showing to be potent (K_A_ 0.9 μM) and quite selective activators of hCA V and VII, respectively.

In an effort to improve both potency and selectivity, in 2022, Chiaramonte et al. developed histamine-related compounds by replacing the imidazole ring with several five-membered heterocycles (pyrrole, N1-substitued imidazole, pyrazole, etc..), and changing the length of the aliphatic chain that leads to the primary amino group [71]. 

According to these modifications, 23 compounds were synthetized and biologically tested on hCA I, II, VA, VII, and XIII isoforms, but only a few of them demonstrated to activate these isoforms better than the lead compound histamine **7** (Table 1). In general, none of the new compounds displayed activity on hCA II, and the most activated isoform was hCA I with K_A_ values in the low micromolar range (0.9–93.5 μM), while the other isoforms were activated with higher K_A_ values (11.2–78.5 μM for hCA V, 9.7–74.6 μM for hCA VII, and from 28.2 to >100 μM for hCA XIII).

Specifically, above all derivatives synthesized, compounds **80**, **81,** and **82** showed an interesting selectivity for activating hCA I over hCA II, VA, VII, and XIII (Table 10). Compounds **80**–**82** could represent lead molecules to obtain new potent and selective hCA I activators [71]. 

**Table 10 molecules-27-02544-t010:** Clonidine (**78**), imidazoline (**79**), and the other related five-membered N-heterocyclic derivatives **80–82** as CAAs.

Compound*n*/Name	Structure	CA Activation	Ref.
**78** **Clonidine**	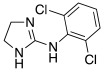	hCA I: K_A_ = 76.3 μMhCA VA: K_A_ = 42.6 μMhCA VII: K_A_ = 8.4 μMhCA XIII: K_A_ = 7.8 μM	[70]
**79**	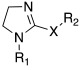 R_1_ = H, CH_3_; R_2_ = alkyl, alkylarylX = NH, NCH_3_, S	hCA I: K_A_ = 4.18- >100 μMhCA VA: K_A_ = 0.9–52.7 μMhCA VII: K_A_ = 0.9–46.7 μMhCA XIII: K_A_ = 6.5- >100 μM	[70]
**79a**	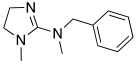	hCA I: K_A_ = 30.2 μMhCA VA: K_A_ = 0.9 μMhCA VII: K_A_ = 6.5 μMhCA XIII: K_A_ = 17.4 μM	[70]
**79b**	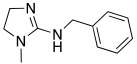	hCA I: K_A_ = 16.9 μMhCA VA: K_A_ = 3.7 μMhCA VII: K_A_ = 0.9 μMhCA XIII: K_A_ = 19.1 μM	[70]
**80**	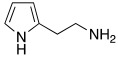	hCA I: K_A_ = 2.16 μMhCA VA: K_A_ = 29.8 μMhCA VII: K_A_ = 44.6 μMhCA XIII: K_A_ > 100 μM	[71]
**81**	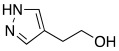	hCA I: K_A_ = 2.19 μMhCA VA: K_A_ = 78.5 μMhCA VII: K_A_ > 100 μMhCA XIII: K_A_ > 100 μM	[71]
**82**	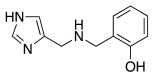	hCA I: K_A_ = 0.9 μMhCA VA: K_A_ = 11.2 μMhCA VII: K_A_ = 13.2 μMhCA XIII: K_A_ > 100 μM	[71]

#### 2.7.4. Indazole, Pyrazole and Oxazole Derivatives Carrying Amino Acidic Tails

In 2016, Maccallini et al. [72] attempted to join the favorable characteristics for activating CAs of heterocyclic compounds such as indazole, pyrazole, and oxazole rings with amino acids characterized by the ability to inhibit neuronal nitric oxide synthase (nNOS), which is an important activity for the treatment of neurodegenerative diseases that are characterized by an abnormal nitrergic signal joined with a low activity and expression of CAs. Thus, in this context, five different series of compounds **83**–**87** (Table 11) that were potentially able to inhibit nNOS and activate CAs were developed. In these series **83–87** (Table 11) a heterocyclic moiety (indazole, pyrazole, and oxazole) coupled with an amino acid residue (Ala, Tyr, and Glu) was present. 

Among all derivatives of series **83**–**87**, five substituted indazoles **85** turned out to inhibit nNOS with high potency and high selectivity with respect to iNOS and eNOS.

Moreover, compounds **83**–**87** were evaluated in vitro on hCA I, II, IV, and VII isozymes, and the obtained results revealed a slight activation of isoforms II and IV in the micromolar range (K_A_ 4.0–43.2 μM and 7.1–48.6 μM, respectively) for all tested compounds. Some derivatives, **84b** (R = *p*-OH-C_6_H_4_), **85a** (R = CH_3_), and **85c** (R = CH_2_CH_2_COOH) activated the hCA VII isoform with K_A_ in the low micromolar range (K_A_ values of 0.69 μM, 0.59 μM, and 0.51 μM, respectively), but without a clear correlation between enzyme activation and chemical structure. The most activated isoform was hCA I with K_A_ values ranging from 9.0 to 6.39 μM for all compounds **83**–**87**: of note, the 1H-indazole series **84** (K_A_ values ranging from 9.0 nM to 1.25 μM), with compound **84b** (R = *p*-OH-C_6_H_4_) being the best hCA I activator (K_A_ 9 nM). 

Anyhow, the most encouraging compound is the indazole derivative **85b** (K_A_ 15 nM towards hCA I), which also displayed the best percentage of nNOS inhibition, thus showing to be a dual agent able to perform a selective nNOS inhibition and hCA I activation. Derivative **85b** is therefore a promising lead compound to develop new drug candidates for the therapy of neurodegeneration [72]. 

#### 2.7.5. Indole-Based Derivatives 

In 2021, Barresi et al. [73] developed three series (**88**–**90**, Table 12) of compounds with an indole central scaffold, which is known to be a “privileged scaffold” for drug discovery, due to its versatility and usefulness [74]. 

The indole-based derivatives **88**–**90** were decorated at N1 with a benzyl group in all series, while different substitution patterns at 3- and 5-positions were exploited. Specifically, in series **88** and **89**, 3-position features a glyoxylamide (**88**) or carboxamide (**89**) moiety in which the N-atom is substituted (R_1_) with polar (hydroxypropyl or hydroxyethyl), protonatable groups (dimethylaminopropyl or diethylaminoethyl) or with a benzyl group; conversely, in series **90** an ethyl ester is present at 3-position of the scaffold. The indole 5-position is substituted with hydroxyethyl or with protonatable groups (dimethylaminopropyl or dimethylaminoethyl or diethylaminoethyl) [73]. 

Nine compounds were synthetized and tested on the main isoforms expressed in human brain, namely hCA I, II, VA, and VII. Isoforms hCA I and II showed not to be significantly activated by all compounds, with K_A_ values ranging between 69.1 μM and values > 100 μM. Isoform hCA VA demonstrated to be quite activated by all the tested compounds with K_A_ values in the range of 24.4–59.8 μM. The cerebral isoform hCA VII proved to be the most sensitive to the activation by all compounds with K_A_ values ranging between 7.2 and 10.8 μM, independently from the position of the basic group, leading to speculation that what matters is its presence, rather than its position at the scaffold. The best hCA VII activators belong to series **89** and **90**, namely compounds **89a**, **90a,** and **90b,** with K_A_ values of 7.5, 7.2, and 8.2 μM, respectively [73]. Due to these results being extremely encouraging, the three compounds were furtherly biologically characterized.

In particular, compounds **89a**, **90a,** and **90b** were first tested for their cytotoxicity on microglial cells by treating human C20 cells with micromolar concentrations of the target compounds and, then, cell viability was evaluated by MTS assay. The data obtained revealed that, while compounds **90a** and **90b** caused a slight reduction in cell viability, compound **89a** did not induce cytotoxicity in the human microglial cell line C20 at any tested concentration.

Given these results, compound **89a** was selected for further determination of its capability to produce BDNF (brain-derived neurotrophic factor) on the same cell line and, interestingly, it showed to induce the production of BDNF as good as microglia [73,75].

Going deeper, there is a link between the process of activating CAs and the process which leads to the release of BDNF, and this link is the correct acidification process. Indeed, it is important for activating CAs because an activator behaves as an additional proton shuttle towards His64 and so it must have a protonatable moiety in its molecular structure; in addition, it is also important for releasing BDNF from secretory granules, since correct acidification is a fundamental step during the release of BDNF [76]. 

Furthermore, **89a** was found to possess the physicochemical parameters suitable for reaching the CNS.

## 3. Conclusions

Carbonic anhydrases activators (CAAs) are still poorly studied compared to carbonic anhydrases inhibitors (CAIs). Indeed, while CAIs are already used in therapy for the treatment of a number of pathologies such as glaucoma, epilepsy, and obesity, CAAs to date, have not shown relevant pharmacological applications and are still under study. 

CAAs are proposed to participate in the proton transfer, acting as additional proton shuttles towards His64 in the rate-limiting step of CA catalytic cycle, that is, the generation of the active hydroxylated state of the enzyme from the inactive acid form. In this respect, CAAs must feature two main structural requirements: to be a small molecule suitable for the active site and to possess at least a protonatable basic moiety with pKa values in the range of 6.5–8.0 for participating in the proton transfer. 

Interest in CAAs has increased thanks to recent findings reporting an involvement of CAs in cognitive and memory disorders, suggesting that the activation of this enzyme may represent a potential effective strategy for the strengthening of synaptic efficacy. In particular, cerebral isoforms of CAs (mainly hCA I, II, VA, and VII) could be a valid target for the development of new agents useful for the treatment of neurodegenerative pathologies such as Alzheimer’s disease in which these isoforms showed to be less expressed. 

Indeed, it was demonstrated that the administration of an activator to experimental animals determined the enhancement of synaptic efficiency and the consolidation of cognition, spatial memory, and learning. 

Based on these findings, a number of CAAs are under study as potential candidates for the development of new drugs for the treatment of disorders in which memory and cognition are impaired. In this context, this paper reviewed the current state of the art concerning CAAs, focusing attention on those compounds that potently and selectively activate CAs, with particular regard to cerebral isoforms, and for which preliminary biological evaluations were carried out in order to evaluate their potential application in therapy. 

Amino acids and amine represent the most investigated compounds in the field of CAs activation. In fact, many of the compounds here described were obtained using histamine or histidine as lead compounds, and involved, thanks to previous SAR studies, the modification of either the heterocyclic ring, the amino portion, or the introduction of portions that could improve the activity and selectivity for the CA isoforms of interest. Some of them showed to strongly activate hCAI, hCA II, and/or hCA VII isoform with activation constant (K_A_) values in the very low nanomolar range.

As a summary, the hCA activation properties of the most interesting compounds reviewed are highlighted in Table 13. 

In conclusion, in the present review we highlighted the peculiar decorations of CAAs, describing their involvement in the proton shuttling in the rate-limiting step of the CA catalytic cycle. Furthermore, this report provides an exciting outlook on existing data that might help the medicinal chemist in the design and development of more efficient and selective new leads for future in vitro and in vivo studies, so that novel drug candidates can be unveiled and introduced onto the market and into the clinical pipeline. 

## Figures and Tables

**Figure 1 molecules-27-02544-f001:**
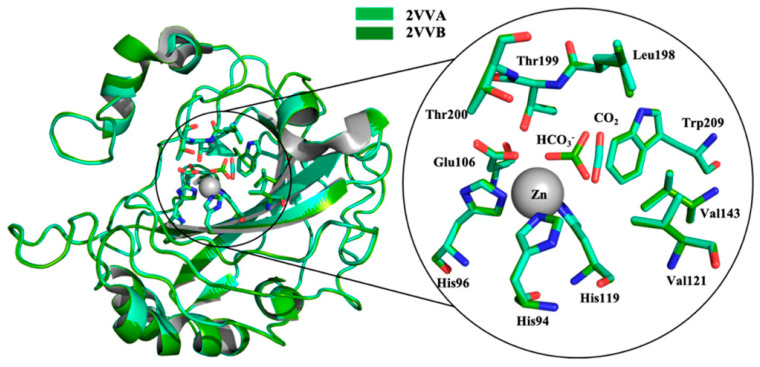
α-CA catalytic site: superposition of CA II crystal structure in complex with CO_2_ (pdb 2VVA) and HCO_3_^−^ (pdb 2VVB). In the circle, an enlargement of the catalytic domain [9]. Under CC BY 4.0 license.

**Figure 2 molecules-27-02544-f002:**
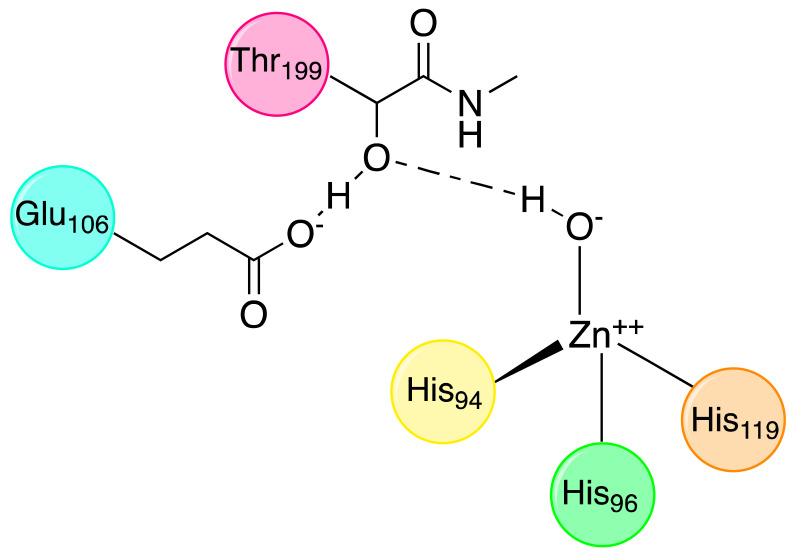
Zinc-bound hydroxylic ion interactions.

**Figure 3 molecules-27-02544-f003:**
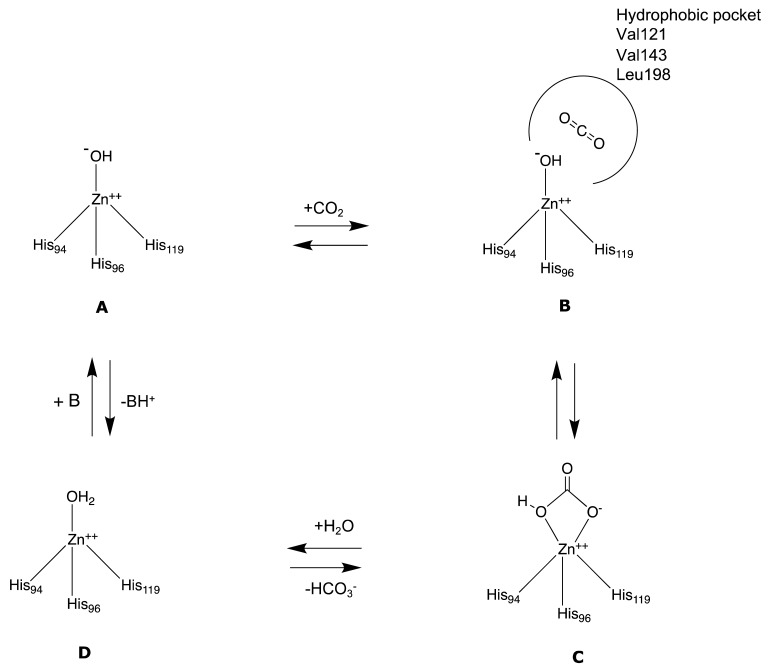
Catalytic cycle of CAs. (**A**)The water molecule transforms itself into a hydroxylated species. (**B**) The hydroxyl ion attacks CO_2_. (**C**) Formation of the species in which HCO_3_^−^ is coordinated to the zinc ion. (**D**) Rebuilding of the inactive state of the enzyme.

**Figure 4 molecules-27-02544-f004:**
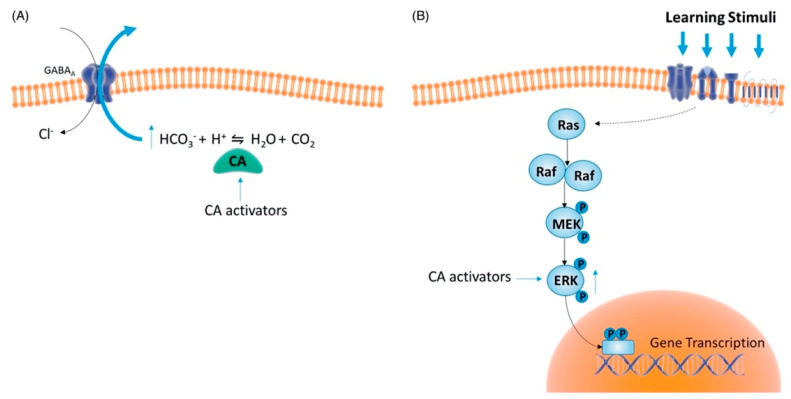
Mechanisms underlying the actions of CAs on cognition [28]: effects on GABAergic post-synaptic potential (**A**); effects on ERK pathways (**B**). Under CC BY 4.0 license.

**Table 1 molecules-27-02544-t001:** Amino acids and amines studied as CAAs.

Compound*n*/Name	Structure	CA Activation	Ref.
**1** **Phenylalanine**	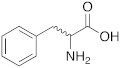	hCA I (L-): K_A_ = 70 nMhCA I (D-): K_A_ = 86 μMhCA II (L-): K_A_ = 13 nMhCA II (D-): K_A_ = 35 nMhCA XIV (L-): K_A_ = 0.24 μMhCA XIV (D-): K_A_ = 7.21 μM	[33]
**2** **Histidine**	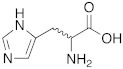	hCA I (L-): K_A_ = 30 nMhCA I (D-): K_A_ = 90 nMhCA VII (L-): K_A_ = 0.92 μΜhCA VII (D-): K_A_ = 0.71 μΜhCA VA (D-): K_A_ = 0.12 μΜ	[34]
**3** **DOPA**	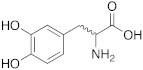	hCA VA (L-): K_A_ = 36 nMhCA VB (L-): K_A_ = 63 nMhCA XIII (D-): K_A_ = 0.81 μM	[4]
**4** **Tryptophan**	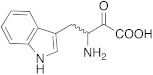	hCA XIII (L-): K_A_ = 0.81 μM	[35]
**5** **Tyrosine**	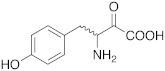		[3,4]
**6** **4-NH_2_-L-Phe**	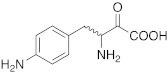		[3]
**7** **Histamine**	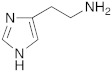	hCA VA: K_A_ = 10 nMhCA XIV: K_A_ = 10 nM	[3,4]
**8** **Dopamine**	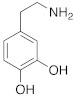	hCA VA: K_A_ = 0.13 μMhCA VII: K_A_ = 0.89 μM	[3,4]
**9** **Serotonine**	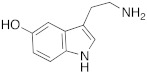		[3,4]
**10***n* = 1**11** *n* = 2	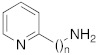		[3,4]
**12** X = NH**13** X = O	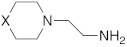	hCA (**13**): K_A_ = 0.13–0.43 μM(except for hCA VB and VII)	[3,4]
**14** **Adrenaline**	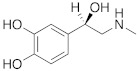	hCA I: K_A_ = 90 nMhCA XII: K_A_ = 0.41 μM	[36]

**Table 2 molecules-27-02544-t002:** CAAs based on histamine structure.

Compound*n*	Structure	CA Activation	Ref.
**15**	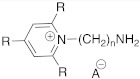	hCA II (R = CH3; *n* = 2):147% activation rate at 10 μM	[38]
**16**	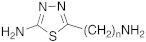	hCA II (*n* = 2):163% activation rate at 10 μM	[38]
**17**	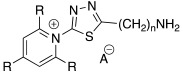 R = alkyl, aryl; *n* = 2,3	hCA II (R = CH3; *n* = 2):184% activation rate at 10 μM	[38]
**18**	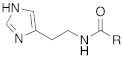	hCA I: K_A_ = 4.0 nM–0.27 μMhCA II: K_A_ = 0.10–0.86 μMbCA IV: K_A_ = 20 nM–21 μM	[39]
**19**	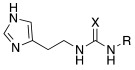 X = O, S	hCA I: K_A_ = 4.0 nM–36 μMhCA II: K_A_ = 80 nM–16 μMbCA IV: K_A_ = 20 nM–12 μMhCA II: K_A_ > 200 μM	[11,39]
**20**	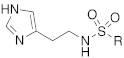	hCA I: K_A_ = 6.0 nM–0.28 μMhCA II: K_A_ = 80 nM–34 μMbCA IV: K_A_ = 10 nM–7.0 μM	[40]
**21**	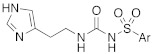	hCA I: K_A_ = 3.0–6.0 nMhCA II: K_A_ = 80 nM–0.15 μMbCA IV: K_A_ = 10–30 nM	[40]
**22**	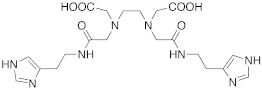	hCA I: K_A_ = 6.0 nMhCA II: K_A_ = 0.12 μMbCA IV: K_A_ = 30 nM	[39]
**23**	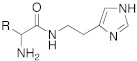	hCA I: K_A_ = 1.0 nM–0.21 μMhCA II: K_A_ = 10 nM–11 μMbCA IV: K_A_ = 3.0 nM–4.6 μM	[41]
**24**	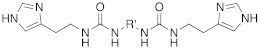	hCA I: K_A_ = 0.73–3.4 μMhCA II: K_A_ > 200 μM	[11]
**24a**	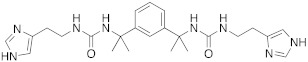	hCA I: K_A_ = 0.73 μM	[11]
**25**	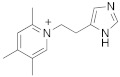	hCA II: 156% activation rate at 20 μM	[42]
**26**	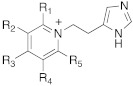	hCA I: K_A_ = 0.5 nM–93 μMhCA II: K_A_ = 9 nM–78 μMhCA VII: K_A_ = 0.8 nM–1.16 μM	[43]
**27**	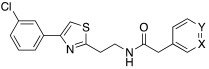 X, Y = CH, N		[44]
**28**	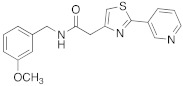	hCA I: K_A_ = 63.4 μMhCA II: K_A_ = 68.1 μMhCA VII: K_A_ = 7.5 μM	[44]

**Table 3 molecules-27-02544-t003:** Histamine based compounds as CAAs.

Compound*n*/Name	Structure	CA Activation	Ref.
**29**	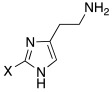 X = Cl, Br, I	hCA I: K_A_ = 0.7–21 nMhCA II: K_A_ = 1.0–115 nM	[45]
**30**	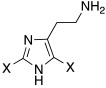 X = Cl, Br, I	hCA I: K_A_ = 0.7–21 nMhCA II: K_A_ = 1.0–115 nM	[45]
**31**	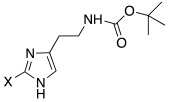 X = Cl, Br, I	hCA I: K_A_ = 5.4–29.3 μMhCA II: K_A_ = 13.6–50.2 μM	[45]
**32**	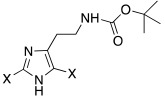 X = Cl, Br, I	hCA I: K_A_ = 5.4–29.3 μMhCA II: K_A_ = 13.6–50.2 μM	[45]
**33**	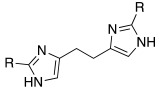 R = H, Me, Et, *i*-Pr, Ph	hCA VA: K_A_ = 9.0–131 nMhCA VII: K_A_ = 15–89 nM	[47]
**34**	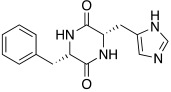		[48]
**35**	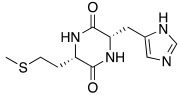		[48]
**36**	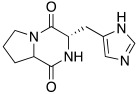		[48]
**37** ***L*-(+)-Ergothioneine**	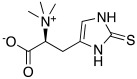		[48]
**38** **Melatonin**	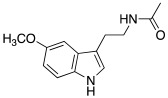		[48]
**39** **Spinacine**	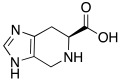		[48]
**40**	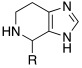 R = Aryl, furyl	hCA VII selective:K_A_ = 82–840 nM	[49]
**41**	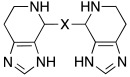 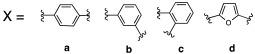	hCA VII selective:K_A_ = 32–39 nM	[50]
**42**	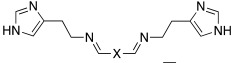 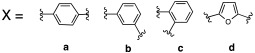	hCA I, II, IV, VII:K_A_ = 3.28–42.1 μMhCA VII (**42d** selective):K_A_ = 85 nM	[50]

**Table 4 molecules-27-02544-t004:** Histidine- and carnosine-based derivatives **43**–**48** as CAAs.

Compound *n*/Name	Structure	CA Activation	Ref.
**43** **Carnosine**	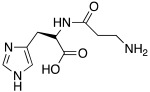		[53]
**44**	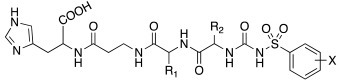 X = 4-F, 2-Me R_1_, R_2_ = amino acidic residues	hCA I, bCA IV: K_A_ = 1.0–20 nM hCA II: K_A_ = 10–40 nM	[53]
**45**	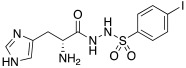	hCA II: K_A_ = 0.21 μM	[54]
**46**	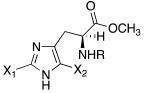 X_1_ = Cl, Br; X_2_ = H, Cl, Br, I R = H, Butyloxycarbonyl (Boc)		[46]
**47**	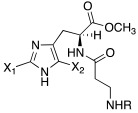 X1 = Cl, Br; X2 = H, Cl, Br, I R = H, Butyloxycarbonyl (Boc)		[46]
**46a**	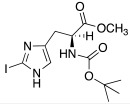	hCA I: K_A_ = 0.9 nM hCA II: K_A_ = 12 nM	[46]
**46b**	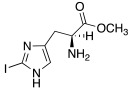	hCA I: K_A_ = 1.2 nM hCA II: K_A_ = 0.32 μM	[46]
**46c**	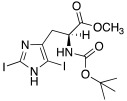	hCA I: K_A_ = 1.5 nM hCA II: K_A_ = 13 nM	[46]
**46d**	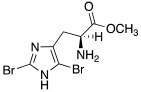	hCA VII: K_A_ = 9.7 μM	[46]
**47a**	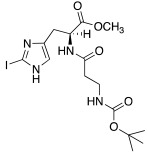	hCA I: K_A_ = 4.3 μM hCA II: K_A_ = 5.2 μM hCA VII: K_A_ = 0.81 μM	[46]
**47b**	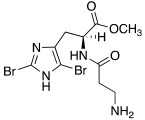	hCA VII: K_A_ = 8.1 nM	[46]
**47c**	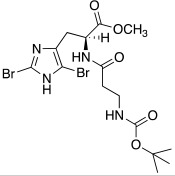	hCA VII: K_A_ = 4.0 μM	[46]
**48**	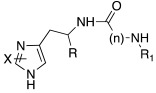 X = H, CH_3_; R_1_ = H, acetyl, Gly; R = H, COOH, CONH_2_, CH_2_OH; *n* = 1, 2, 3		[55]
**48a** **D-Carnosinamide**	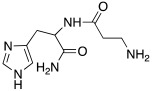	hCA I: K_A_ = 16.6 μM	[55]
**48b** **Carnicine**	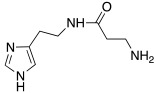	hCA I: K_A_ = 16.6 μM hCA VA: K_A_ = 6.4 μM	[55]
**48c** **L-Anserine**	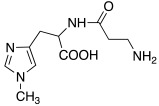	hCA IX: K_A_ = 1.14 μM	[55]

**Table 5 molecules-27-02544-t005:** Gold nanoparticles of histamine, histidine, and carnosine derivatives **49**–**51**, and sulfur-, selenium- and tellurium-containing amines **52**–**54** as CAAs.

Compound *n*	Structure	CA Activation	Ref.
**49**	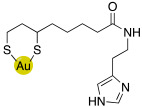	K_A_ = 1–7 nM (details in Table 6)	[57]
**50**	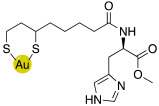	K_A_ = 1–8 nM (details in Table 6)	[57]
**51**	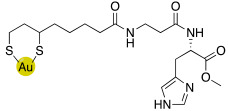	K_A_ = 1–9 nM (details in Table 6)	[57]
**52**	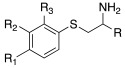	hCA I: K_A_ = 7.7–13.5 μM hCA VA: K_A_ = 10.2–43.7 μM hCA VII: K_A_ = 11.4–23.4 μM	[58]
**53**	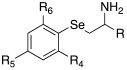	hCA I: K_A_ = 5.2–22.1 μM hCA VA: K_A_ = 6.6–20.9 μM hCA VII: K_A_ = 10.1–23.3 μM	[58]
**54**	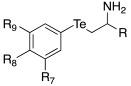	hCA I: K_A_ = 4.6–13.3 μM hCA VA: K_A_ = 3.3–20.2 μM hCA VII: K_A_ = 8.9–44.9 μM	[58]

**Table 6 molecules-27-02544-t006:** Activation data of human CA isoforms I, II, VA, and VII with compounds **49**–**51** in comparison with their starting compounds, using a stopped flow CO_2_ hydrase assay [37].

Isoform/ Compound	K_A_ (μM)					
	hCA I	hCA II	hCA IV	hCA VA	hCA VII	hCA XIV
**Histamine (7)**	2.1	125	25.3	0.010	37.5	0.010
**49**	0.005	0.002	0.001	0.001	0.003	0.007
**L-His (2)**	0.03	10.9	7.3	1.34	0.92	0.90
**L-His-OMe**	0.02	10.4	6.8	1.86	0.88	0.93
**50**	0.002	0.008	0.001	0.002	0.003	0.001
**L-carnosine (43)**	1.1	33	19	1.54	0.75	0.64
**L-carnosine-OMe**	10.9	32	18	1.36	0.84	0.71
**51**	0.009	0.007	0.002	0.002	0.001	0.001

**Table 7 molecules-27-02544-t007:** Structures of drugs **55–64** as CAAs.

Compound *n*/Name	Structure	CA Activation	Ref.
**55** **Timolol**	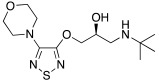	hCA I: K_A_ = 12 μM hCA II: K_A_ = 9.3 μM	[60]
**56** **Fluoxetine**	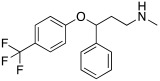	hCA I: 175% activation rate at 1 μM hCA II: 165% activation rate at 1 μM	[61]
**57** **Sertraline**	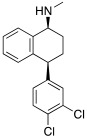	hCA I: 145% activation rate at 1 μM hCA II: 140% activation rate at 1 μM	[61]
**58** **Citalopram**	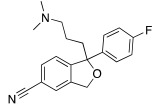	hCA I: 134% activation rate at 1 μM hCA II: 170% activation rate at 1 μM	[61]
**59** **Sildenafil**	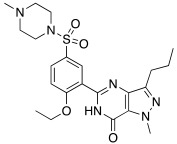	hCA I: K_A_ = 1.08 μM hCA VB: K_A_ = 6.54 μM hCA VI: K_A_ = 2.37 μM	[62]
**60** **Amphetamine**	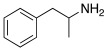	hCA IV: K_A_ = 94 nM hCA VA: K_A_ = 0.81 μM hCA VB: K_A_ = 2.56 μM hCA VII: K_A_ = 0.91 μM	[59]
**61** **Methamphetamine**	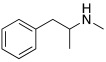	hCA IV: K_A_ = 51 nM hCA VA: K_A_ = 0.92 μM hCA VB: K_A_ = 0.78 μM hCA VII: K_A_ = 0.93 μM	[59]
**62** **Phentermine**	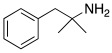	hCA IV: K_A_ = 74 nM hCA VA: K_A_ = 0.53 μM hCA VB: K_A_ = 0.62 μM hCA VII: K_A_ = 0.89 μM	[59]
**63** **Mephentermine**	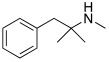	hCA IV: K_A_ = 1.03 μM hCA VA: K_A_ = 0.37 μM hCA VB: K_A_ = 0.24 μM hCA VII: K_A_ = 0.64 μM	[59]
**64** **Chlorphenteramine**	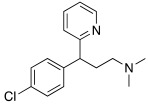	hCA IV: K_A_ = 55 nM hCA VA: K_A_ = 0.31 μM hCA VB: K_A_ = 0.75 μM hCA VII: K_A_ = 98 nM	[59]

**Table 8 molecules-27-02544-t008:** Structures of histamine receptors agonists/antagonists **65–74** as CAAs.

Compound*n*/Name	Structure	CA Activation	Ref.
**65** **α-methyl histamine**	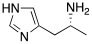	hCA I: K_A_ = 0.12 μMhCA II: K_A_ = 82 nMhCA VII: K_A_ = 1.25 μM	[66]
**66** **4-methyl histamine**	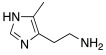	hCA I: K_A_ = 0.36 μMhCA II: K_A_ = 5.4 μMhCA VII: K_A_ = 0.39 μM	[66]
**67** **1-methyl histamine**	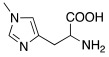	hCA I: K_A_ = 52 nMhCA II: K_A_ = 0.57 μMhCA VII: K_A_ = 0.19 μM	[66]
**68** **2-(2-aminoethyl) thiazole**	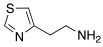	hCA I: K_A_ = 0.87 μMhCA II: K_A_ = 7.45 μMhCA VII: K_A_ = 0.7 μM	[66]
**69** **Burimamide**	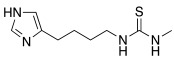	hCA I: K_A_ = 0.88 μMhCA II: K_A_ = 8.39 μMhCA VII: K_A_ = 0.43 μM	[66]
**70** **Metiamide**	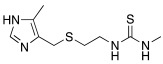	hCA I: K_A_ = 0.98 μMhCA II: K_A_ = 8.75 μMhCA VII: K_A_ = 1.01 μM	[66]
**71** **Impromidine**	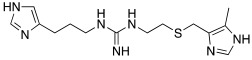	hCA I: K_A_ = 0.72 μMhCA II: K_A_ = 2.14 μMhCA VII: K_A_ = 0.10 μM	[66]
**72** **Methimmepip**	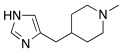	hCA I: K_A_ = 3.16 μMhCA II: K_A_ = 5.24 μMhCA VII: K_A_ = 0.12 μM	[66]
**73** **Proxyfan**	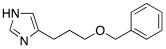	hCA I: K_A_ = 3.15 μMhCA II: K_A_ = 7.66 μMhCA VII: K_A_ = 0.52 μM	[66]
**74** **Ciproxyfan**	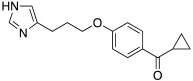	hCA I: K_A_ = 4.29 μMhCA II: K_A_ = 9.9 μMhCA VII: K_A_ = 0.11 μM	[66]

**Table 9 molecules-27-02544-t009:** Ureas and di-ureas incorporating 1,2,4-triazole derivatives **75**–**76** and amino alcohol oxime ethers **77** as CAAs.

Compound *n*	Structure	CA Activation	Ref.
**75**	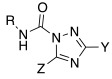 R, R′ = alkyl, aryl Y = NH_2_, H Z = H, NH_2_, COOH		[67]
**76**	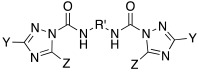 R, R′ = alkyl, aryl Y = NH_2_, H Z = H, NH_2_, COOH		[67]
**75a**	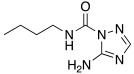	hCA I: K_A_ = 6.1 nM hCA II: K_A_ = 1.7 nM	[67]
**76a**	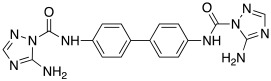	hCA I: K_A_ = 0.81 nM hCA II: K_A_ = 14.4 nM	[67]
**76b**	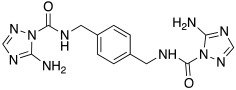	hCA I: K_A_ = 0.94 nM hCA II: K_A_ = 0.05 nM	[67]
**76c**	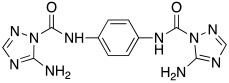	hCA I: K_A_ = 65 nM hCA II: K_A_ = 0.12 nM	[67]
**77**	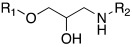 R_1_ = aryl, cicloalkyl R_2_ = *i*-propyl, *t*-butyl		[68]
**77a**	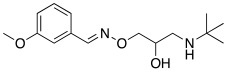	hCA I: K_A_ = 7.10 μM hCA II: K_A_ = 79 nM hCA IV: K_A_ = 6.01 μM hCA VII: K_A_ = 0.42 μM	[68]
**77b**	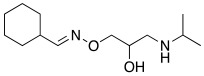	hCA I: K_A_ = 12.1 μM hCA II: K_A_ = 2.50 μM hCA IV: K_A_ = 7.73 μM hCA VII: K_A_ = 82 nM	[68]
**77c**	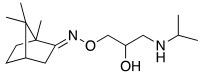	hCA I: K_A_ = 8.15 μM hCA II: K_A_ = 1.94 μM hCA IV: K_A_ = 1.08 μM hCA VII: K_A_ = 91 nM	[68]

**Table 11 molecules-27-02544-t011:** Indazole, pyrazole, and oxazole derivatives **83**–**87** as CAAs.

Compound *n*	Structure	CA Activation	Ref.
**83a–c**	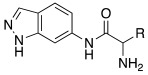 **a:** R = CH_3_ (Ala) **b:** R = *p*-OH-C_6_H_4_ (Tyr) **c:** R = CH_2_CH_2_COOH (Glu)		[72]
**84a–c**	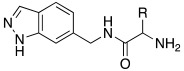 **a:** R = CH_3_ (Ala) **b:** R = *p*-OH-C_6_H_4_ (Tyr) **c:** R = CH_2_CH_2_COOH (Glu)		[72]
**85a–c**	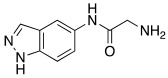 **a:** R = CH_3_ (Ala) **b:** R = *p*-OH-C_6_H_4_ (Tyr) **c:** R = CH_2_CH_2_COOH (Glu)		[72]
**86a–c**	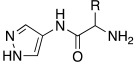 **a:** R = CH_3_ (Ala) **b:** R = *p*-OH-C_6_H_4_ (Tyr) **c:** R = CH_2_CH_2_COOH (Glu)		[72]
**87a–c**	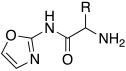 **a:** R = CH_3_ (Ala) **b:** R = *p*-OH-C_6_H_4_ (Tyr) **c:** R = CH_2_CH_2_COOH (Glu)		[72]
**84b**	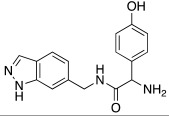	hCA I: K_A_ = 9.0 nM hCA VII: K_A_ = 0.69 μM	[72]
**85a**	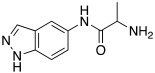	hCA I: K_A_ = 6.39 nM hCA VII: K_A_ = 0.59 μM	[72]
**85b**	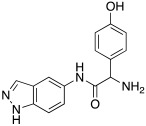	hCA I: K_A_ = 15 nM hCA VII: K_A_ = 10.8 μM	[72]
**85c**	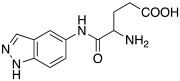	hCA I: K_A_ = 4.12 μM hCA VII: K_A_ = 0.51 μM	[72]

**Table 12 molecules-27-02544-t012:** Indole-based derivatives **88**–**90** as CAAs.

Compound *n*/	Structure	CA Activation	Ref.
**88**	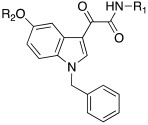 R_1_ = CH_2_CH_2_OH, CH_2_CH_2_CH_2_OH, CH_2_CH_2_N(CH_3_)_2_, CH_2_CH_2_N(C_2_H_5_)_2_, CH_2_CH_2_CH_2_N(CH3)_2_, CH_2_Ph R_2_ = CH_2_CH_2_OH, CH_2_CH_2_N(CH_3_)_2_, CH_2_CH_2_N(C_2_H_5_)_2_, CH_2_CH_2_CH_2_N(CH_3_)_2_		[73]
**89**	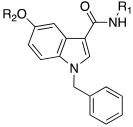 R_1_ = CH_2_CH_2_OH, CH_2_CH_2_CH_2_OH, CH_2_CH_2_N(CH_3_)_2_, CH_2_CH_2_N(C_2_H_5_)_2_, CH_2_CH_2_CH_2_N(CH3)_2_, CH_2_Ph R_2_ = CH_2_CH_2_OH, CH_2_CH_2_N(CH_3_)_2_, CH_2_CH_2_N(C_2_H_5_)_2_, CH_2_CH_2_CH_2_N(CH_3_)_2_		[73]
**90**	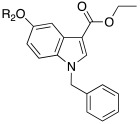 R_1_ = CH_2_CH_2_OH, CH_2_CH_2_CH_2_OH, CH_2_CH_2_N(CH_3_)_2_, CH_2_CH_2_N(C_2_H_5_)_2_, CH_2_CH_2_CH_2_N(CH3)_2_, CH_2_Ph R_2_ = CH_2_CH_2_OH, CH_2_CH_2_N(CH_3_)_2_, CH_2_CH_2_N(C_2_H_5_)_2_, CH_2_CH_2_CH_2_N(CH_3_)_2_		[73]
**89a**	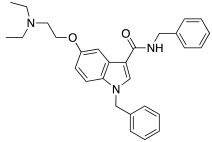	hCA VII: K_A_ = 7.5 μM	[73]
**90a**	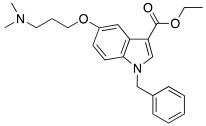	hCA VII: K_A_ = 7.2 μM	[73]
**90b**	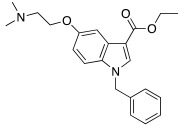	hCA VII: K_A_ = 8.2 μM	[73]

**Table 13 molecules-27-02544-t013:** Properties of the most interesting compounds reviewed.

Compound	Class	In Vitro Activity		Other Correlate Activities	Ref.
		hCA Activation Activity (K_A_)	Selectivity		
**21**	Arylsulfonylureido derivatives of histamine	3–6 nM	hCA I	-	[40]
**22**	Histamine dimers	6 nM	hCA I	-	[39]
**29–30**	Histamine-based halogenated compounds	0.7–21 nM 1.0–115 nM	dual hCA I/ hCA II	-	[45]
**41**	Histamine inspired-compounds	32–39 nM	hCA VII	-	[49]
**42d**	Histamine inspired-compounds	85 nM	hCA VII	-	[49]
**44**	Carnosine-based derivatives	1–20 nM 10–40 nM	dual hCA I/ hCA II	Enhancement of red cell CA activity	[53]
**46a**	Histidine-based derivatives	0.9 nM	hCA I	-	[46]
**47b**	Carnosine-based derivatives	8.1 nM	hCA VII	-	[46]
**49–51**	Gold nanoparticles of histamine, histidine, carnosine derivatives	1–9 nM	no-selectivity (I, II, IV, VA, VII, IVX)	-	[45]
**53–54**	Selenium and tellurium containing amines	3.3–44.9 μM	no-selectivity (I, VA, VII)	ROS inhibition	[58]
**76b**	Di-ureas incorporating 1,2,4-triazole derivatives	0.05 nM	hCA II	-	[67]
**77a**	Amino alcohol oxime ethers	79 nM	hCA II	-	[68]
**77b**	Amino alcohol oxime ethers	82 nM	hCA VII	-	[68]
**85b**	Indazole derivartives	15 nM	hCA I	nNOS inhibitor	[72]
**89a**	Indole-based derivatives	7.5 μM	hCA VII	BDNF production	[73]

## Data Availability

Data are contained within the article.

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
