# Peer review of "Carbonic Anhydrase Activators for Neurodegeneration: An Overview"

_molecules, 2022, doi:10.3390/molecules27082544_

Round 1

Reviewer 1 Report

The review entitled “carbonic anhydrase activators for neurodegeneration: an overview” describes the research about the compounds endowed with the ability to activate CA isoforms involved in neurodegenerative disorders.

The review is well written and the different scaffolds have been properly described. SAR analysis have been also reported when available.

I suggest its acceptance with minor revision.

I also suggest the following minor changes:

Page 1, line 29. Please change “more specifically zinc enzymes”. Some classes possess other metals.

Page 1, lines 33-36. Please correct this sentence by adding all the classes.

Page 2, line 52. Please change “processes” with another word (for example functions).

Page 3, lines 74-76. Please revise this sentence. The interactions are referred to water.

Page 3, lines 77-79. Please revise this sentence.

Page 3, line 72. Please change the term “activation” (it would be convenient to use it for CAAs).

Page 4, line 117. Please remove “supporting this theory”. In my opinion could be useful to briefly cite in this part the importance of His64 for its proton-shuttling role.

Page 6, line 200. Please remove “that”.

Page 6, line 213. Please change “exist” with “occur”.

Page 10, line 323. Please add the point.

Page 10, Figure 6. Please add X and Y for compounds 27.

Page 11, line 337. Please remove “also”.

Page 11, lines 343-344.  This sentence should be argued.

Page 11, lines 351-354. In this sentence is assessed that ureido 19 did not activate hCAII. This aspect should be clarified in the Figure 6.

Page 12, line 400. Please correct “N-butylossicarbonyl”.

Page 14, line 428. Please correct “activator”.

Page 16, line 495. Please remove performing.

Page 17, line 552. Please correct “b-aminochalcogenides”.

Page 21, line 672. Please add “from” after resulting.

Page 21, line 680. Please add comma after triazole.

Page 23, line 718. Please change “is” with “was”.

Author Response

Reviewer 1:

  1. Page 1, line 29. Please change “more specifically zinc enzymes”. Some classes possess other metals.

Response 1: We thank the reviewer for the suggestion. The sentence “more specifically zinc enzymes” has been deleted. p. 1, line 29 of the revised version.

  1. Page 1, lines 33-36. Please correct this sentence by adding all the classes.

Response 2: The sentence has been corrected and all classes have been added. Moreover, an appropriate reference ([5]) has been inserted. p. 1, lines 33-38 of the revised version.

  1. Page 2, line 52. Please change “processes” with another word (for example functions).

Response 3: The word “processes” has been changed with “functions”. p. 2, line 54 of the revised version.

  1. Page 3, lines 74-76. Please revise this sentence. The interactions are referred to water.

Response 4: We thank the reviewer for the suggestion. The sentence “These interactions direct the substrate, represented by CO2, in a favourable position for the nucleophilic attack by water, which loses a proton and increases its nucleophilicity“ has been changed in: “These interactions direct the substrate, represented by CO2, in a favourable position for the nucleophilic attack by the zinc-bound hydroxide, deriving from water which loses a proton and increases its nucleophilicity“. p. 3, lines 76-78 of the revised version.

  1. Page 3, lines 77-79. Please revise this sentence.

Response 5: The sentence “During the process of activation, the water molecule loses a proton, and it transforms itself into a hydroxylated species” has been changed in “In the first step, the water molecule loses a proton, and transforms itself into a hydroxylated species”. p. 3, lines 79-81 of the revised version.

  1. Page 3, line 72 (maybe 92). Please change the term “activation” (it would be convenient to use it for CAAs).

Response 6: The term “activation process” has been changed with “catalytic cycle of the enzyme”. p. 3, line 94 of the revised version.

  1. Page 4, line 117. Please remove “supporting this theory”. In my opinion could be useful to briefly cite in this part the importance of His64 for its proton-shuttling role.

Response 7: “Supporting this theory” has been removed. Moreover, the importance of His64 for its proton-shuttling role has been briefly discussed. p. 4, lines 120-121 of the revised version.

  1. Page 6, line 200. Please remove “that”.

Response 8: As suggested by the reviewer “that” has been removed. p. 6, line 206 of the revised version.

  1. Page 6, line 213. Please change “exist” with “occur”.

Response 9: The term “exist” has been changed with “occur”. p. 6, line 219 of the revised version.

  1. Page 10, line 323. Please add the point.

Response 10: The point is added. p. 11, line 330 of the revised version.

  1. Page 10, Figure 6. Please add X and Y for compounds 27.

Response 11: X, Y = CH or N have been added for compounds 27. p. 10, Table 2 of the revised version.

  1. Page 11, line 337. Please remove “also”.

Response 12: As suggested by the reviewer “also” has been removed. p. 11, line 344 of the revised version.

  1. Page 11, lines 343-344. This sentence should be argued.

Response 13: We thank the reviewer for the suggestion. The sentence has been modified in “These series of CAAs might lead to the development of drugs/diagnostic agents for genetic disease of bone, kidneys and above all, brain, since they mainly activate the CA isoforms most expressed in these compartments (namely hCA I, hCA II, hCA VII). Moreover, some appropriate references [39-41] have been again mentioned here. p. 11, lines 350-352 of the revised version.

  1. Page 11, lines 351-354. In this sentence it is assessed that ureido 19 did not activate hCAII. This aspect should be clarified in the Figure 6.

Response 14: We agree with the reviewer. Actually, general formula 19 refers to two different series of compounds developed in two different studies by Supuran 2000 e Licsandru 2017. Derivatives from Supuran library activate CA II with KA values of 80nM-16mM, whereas derivatives by Licsandru are inactive versus CA II (KA values > 200 mM). In the revised version, this issue has been clarified as you can see in Table 2 that, in this version, has replaced Figure 6.

  1. Page 12, line 400. Please correct “N-butylossicarbonyl”.

Response 15: As rightly suggested by the reviewer “N-butylossicarbonyl” has been corrected. p. 12, line 408 of the revised version.

  1. Page 14, line 428. Please correct “activator”.

Response 16: The term “activator” has been corrected. p. 14, line 436 of the revised version.

  1. Page 16, line 495. Please remove performing.

Response 17: The term “performing” has been removed. p. 18, line 503 of the revised version.

  1. Page 17, line 552. Please correct “b-aminochalcogenides”.

Response 18: “b-aminochalcogenides” has been corrected. p. 20, line 558 of the revised version.

  1. Page 21, line 672. Please add “from” after resulting.

Response 19: As suggested by the reviewer “from” has been added after “resulting”. p. 23, line 678 of the revised version.

  1. Page 21, line 680. Please add comma after triazole.

Response 20: A comma has been added after “triazole”. p. 24, line 686 of the revised version.

  1. Page 23, line 718. Please change “is” with “was”.

Response 21: The term “is” has been changed with “was”. p. 26, line 723 of the revised version.

Reviewer 2 Report

The authors presented a research review on “Carbonic Anhydrase activators for neurodegeneration: an overview”. The review is written well, informative and suitable to be published in the journal after major revision.

Comments and Suggestions for Authors

  1. Better to convert the structures in figure 5-14 with carbonic anhydrase values and citation in the tabulated form having structures, numbering, activity and citation as well.
  2. The section 1.1 “CARBONIC ANHYDRASES ACTIVATION” must be in small letter
  3. The section 1.2. “POTENTIAL THERAPEUTIC APPLICATIONS OF CAAs” must be in small letter.
  4. The section 2. CARBONIC ANHYDRASE ACTIVATORS (CAAS) must be in small letter.
  5. The authors did not cite some recently published work related to the same study. Better to cite the recently published papers on Carbonic anhydrase.

Lemon N, Canepa E, Ilies MA, Fossati S. Carbonic Anhydrases as Potential Targets Against Neurovascular Unit Dysfunction in Alzheimer’s Disease and Stroke. Frontiers in Aging Neuroscience. 2021:785.

Rafiq K, Ur Rehman N, Halim SA, Khan M, Khan A, Al-Harrasi A. Design, Synthesis and Molecular Docking Study of Novel 3-Phenyl-β-Alanine-Based Oxadiazole Analogues as Potent Carbonic Anhydrase II Inhibitors. Molecules. 2022 Jan 26;27(3):816.

Avula SK, Rehman NU, Khan M, Halim SA, Khan A, Rafiq K, Csuk R, Das B, Al-Harrasi A. New synthetic 1H-1, 2, 3-triazole derivatives of 3-O-acetyl-β-boswellic acid and 3-O-acetyl-11-keto-β-boswellic acid from Boswellia sacra inhibit carbonic anhydrase II in vitro. Medicinal Chemistry Research. 2021 Jun;30(6):1185-98.

Rafiq K, Khan A, Ur Rehman N, Halim SA, Khan M, Ali L, Hilal Al-Balushi A, Al-Busaidi HK, Al-Harrasi A. New Carbonic Anhydrase-II Inhibitors from Marine Macro Brown Alga Dictyopteris hoytii Supported by In Silico Studies. Molecules. 2021 Jan;26(23):7074.

  1. I have seen recently published review on the same topic. What is the additional work presented in the current review?

Supuran CT. Carbonic anhydrase activators. Future medicinal chemistry. 2018 Mar;10(5):561-73.

Amine- and Amino Acid-Based Compounds as Carbonic Anhydrase Activators.

Pollard A, Shephard F, Freed J, Liddell S, Chakrabarti L. Mitochondrial proteomic profiling reveals increased carbonic anhydrase II in aging and neurodegeneration. Aging (Albany NY). 2016 Oct;8(10):2425.

  1. The references need to be double check to avoid any redundancy and repetition.

Author Response

Reviewer 2 :

  1. Better to convert the structures in figure 5-14 with carbonic anhydrase values and citation in the tabulated form having structures, numbering, activity and citation as well.

Response 1: We thank the reviewer for the suggestion. The structures in figure 5-14 with carbonic anhydrase values and citation have been converted in the tabulated form (Tables 1-5 and 7-12).

  1. The section 1.1 “CARBONIC ANHYDRASES ACTIVATION” must be in small letter

Response 2: As rightly suggested by the reviewer the section 1.1 “CARBONIC ANHYDRASES ACTIVATION” has been written in lowercase letters in the revised manuscript.

  1. The section 1.2. “POTENTIAL THERAPEUTIC APPLICATIONS OF CAAs” must be in small letter.

Response 3: The section 1.2. “POTENTIAL THERAPEUTIC APPLICATIONS OF CAAs” has been written in lowercase letters in the revised manuscript.

  1. The section 2. CARBONIC ANHYDRASE ACTIVATORS (CAAS) must be in small letter.

Response 4: The section 2. CARBONIC ANHYDRASE ACTIVATORS (CAAS) has been written in lowercase letters in the revised manuscript.

  1. The authors did not cite some recently published work related to the same study. Better to cite the recently published papers on Carbonic anhydrase.

Lemon N, Canepa E, Ilies MA, Fossati S. Carbonic Anhydrases as Potential Targets Against Neurovascular Unit Dysfunction in Alzheimer’s Disease and Stroke. Frontiers in Aging Neuroscience. 2021:785.

Rafiq K, Ur Rehman N, Halim SA, Khan M, Khan A, Al-Harrasi A. Design, Synthesis and Molecular Docking Study of Novel 3-Phenyl-β-Alanine-Based Oxadiazole Analogues as Potent Carbonic Anhydrase II Inhibitors. Molecules. 2022 Jan 26;27(3):816.

Avula SK, Rehman NU, Khan M, Halim SA, Khan A, Rafiq K, Csuk R, Das B, Al-Harrasi A. New synthetic 1H-1, 2, 3-triazole derivatives of 3-O-acetyl-β-boswellic acid and 3-O-acetyl-11-keto-β-boswellic acid from Boswellia sacra inhibit carbonic anhydrase II in vitro. Medicinal Chemistry Research. 2021 Jun;30(6):1185-98.

Rafiq K, Khan A, Ur Rehman N, Halim SA, Khan M, Ali L, Hilal Al-Balushi A, Al-Busaidi HK, Al-Harrasi A. New Carbonic Anhydrase-II Inhibitors from Marine Macro Brown Alga Dictyopteris hoytii Supported by In Silico Studies. Molecules. 2021 Jan;26(23):7074.

Response 5: We thank the reviewer for the suggestion. However, we take the liberty to outline that our review aims to furnish to the reader an overview on the more recently developed CAAs, with particular attention to those with potential for the treatment of neurodegenerative diseases. In this respect, the references (Avula et al. 2021, Rafiq et al. 2021 and 2022) proposed by the reviewer fall out of the topic of our manuscript as they report about studies on novel carbonic anhydrase inhibitors. On the contrary, the review “Lemon et al. S Carbonic Anhydrases as Potential Targets Against Neurovascular Unit Dysfunction in Alzheimer’s Disease and Stroke. Frontiers in Aging Neuroscience. 2021:7852” that highlights the impact of CA modulation in neurological disorders and introduces the idea of potentially repurposing CAIs for prevention of cerebrovascular and neurovascular pathology in AD and stroke, has been briefly reported in the text and cited as reference [21] in the 1.2. section.

Furthermore, during the revision process, this section has been further updated citing a 2022 article about carbonic anhydrases activity in the social recognition memory in rats, reference [16] in the 1.2. section.

[16]     Schmidt, S.D.; Nachtigall, E.G.; Marcondes, L.A.; Zanluchi, A.; Furini, C.R.G.; Passani, M.B.; Supuran, C.T.; Blandina, P.; Izquierdo, I.; Provensi, G.; de Carvalho Myskiw, J. Modulation of Carbonic Anhydrases Activity in the Hippocampus or Prefrontal Cortex Differentially Affects Social Recognition Memory in Rats. Neuroscience, 2022.

  1. I have seen recently published review on the same topic. What is the additional work presented in the current review?

Supuran CT. Carbonic anhydrase activators. Future medicinal chemistry. 2018 Mar;10(5):561-73.

Amine- and Amino Acid-Based Compounds as Carbonic Anhydrase Activators.

Pollard A, Shephard F, Freed J, Liddell S, Chakrabarti L. Mitochondrial proteomic profiling reveals increased carbonic anhydrase II in aging and neurodegeneration. Aging (Albany NY). 2016 Oct;8(10):2425.

Response 6: In the current review, the anhydrase activators (CAAs) already described in the previous reviews cited by the reviewer, are briefly summarized and the newest compounds developed after 2018 and not reported before, are instead widely described. In addition, compared to previous reviews, the present one has been focused on those CAAs that selectively activate CA involved in brain processes and that, therefore, could be developed for treating neurodegenerative diseases. This issue has been better clarified in the revised version of the manuscript at the beginning of Chapter 2. p. 7, lines 243-245.

  1. The references need to be double check to avoid any redundancy and repetition.

Response 7:As suggested by the reviewer all the references have been double checked.

Round 2

Reviewer 2 Report

The manuscript has been revised properly and suitable for publication